# Over-reliance on land for carbon dioxide removal in net-zero climate pledges

Kate Dooley [1], Kirstine Lund Christiansen [2] ✉, Jens Friis Lund [2], Wim Carton [3] & Alister Self [1,4]

Achieving net-zero climate targets requires some level of carbon dioxide removal. Current assessments focus on tonnes of $CO_2$ removed, without specifying what form these removals will take. Here, we show that countries' climate pledges require approximately 1 (0.9–1.1) billion ha of land for removals. For over 40% of this area, the pledges envisage the conversion of existing land uses to forests, while the remaining area restores existing ecosystems and land uses. We analyse how this demand for land is distributed geographically and over time. The results are concerning, both in terms of the aggregate area of land, but also the rate and extent of land use change. Our findings demonstrate a gap between governments' expected reliance on land and the role that land can realistically play in climate mitigation. This adds another layer to the observed shortcomings of national climate pledges and indicates a need for more transparency around the role of land in national climate mitigation plans.

Achieving global net zero carbon dioxide ($CO_2$) emissions is necessary to halt warming on multidecadal timescales[1]. Some level of carbon dioxide removal (CDR) will be required to balance residual emissions at the point of net zero, though both the scale of CDR and the amount of residual emissions remain contested[2]. Multiple studies now assess the current and pledged level of CDR in government and corporate climate pledges for net zero[3–5]. Yet these assessments focus on quantifying removals in terms of tonnes of $CO_2$, which leaves open the question of what form these removals will take. Currently, CDR is generated almost exclusively through land-based measures, and an important question is what land requirements follow from this approach and how this might conflict with other land uses.

The Paris Agreement's goal of achieving a balance between anthropogenic emissions by sources and removals by sinks of greenhouse gasses in the second half of this century[6] has put a spotlight on the role of forests and land in climate mitigation, given the potential to enhance sequestration of atmospheric carbon. Land is central to addressing the accelerating and entwined crises of climate change and biodiversity loss, yet the many and often competing demands made on land are resulting in increasing pressures[7]. Land-use change is

currently the leading driver of biodiversity loss, as well as contributing to climate change, which in turn is expected to drive further biodiversity loss[8]. CDR efforts through land-based approaches, such as large-scale afforestation and bioenergy carbon capture and storage (BECCS) could reduce future biodiversity losses indirectly, by contributing to climate mitigation[9]. Such initiatives may also exacerbate the biodiversity crisis due to additional demand for land and other resources[10–12], with negative impacts on biodiversity concentrated in regions with high land-use change[13]. Land-based CDR can also risk undermining the livelihoods of Indigenous Peoples and other vulnerable and land-dependent communities by dispossessing them of access to land-based resources[14,15].

The centrality of land to CDR options in both modelled scenarios and current policies gives rise to growing uncertainties about the potential aggregate demand for land to address climate mitigation[16]. Under the Paris Agreement, countries make climate mitigation, adaptation and finance pledges in the form of Nationally Determined Contributions (NDCs) every five years. Countries have also agreed to submit Long Term and Low Emissions Development Strategies (LT-LEDS) containing mid-century climate pledges, which are increasingly framed as

[1]School of Geography, Earth and Atmospheric Sciences, The University of Melbourne, Parkville, VIC, Australia. [2]Department of Food and Resource Economics, University of Copenhagen, Frederiksberg C, Denmark. [3]Centre for Sustainability Studies, Lund University, Lund, Sweden. [4]Climate Resource, Melbourne, VIC, Australia. ✉e-mail: klc@ifro.ku.dk

net zero goals. Recent assessments have found both the NDCs and 2050 pledges to be broadly inadequate in meeting the goals of the Paris Agreement[17–19]. Adding to these concerns over the level of ambition in national mitigation pledges, assessments show many countries plan to rely on substantial levels of CDR, including enhanced land carbon sinks[20,21].

While these pledges do not directly translate to national policy priorities and can be interpreted more as aspirational targets, they still give a reasonable indication of the long-term strategies that countries envision. Moreover, even a discursive reliance on CDR can have political implications in the present, by triggering what some have termed a 'spiral of delay' where vague statements of future ambition displace near-term mitigation commitments[22]. This underscores the importance of examining the extent of land implied in the envisaged removals. This matters both to the transparency and integrity of the climate pledges in isolation, but also to their potential impacts on other societal priorities, such as food production and biodiversity protection.

In this study, we assess the expected reliance on land in the mitigation component of national pledges, by reviewing all LT-LEDS or NDCs submitted to the UNFCCC up until the end of 2023 (see methods for more information), building on previous work in the Land Gap report[23]. Our analysis provides insights into the types of land activities that are envisioned for climate mitigation, and how these are distributed geographically and over time. While the information given by countries in their climate pledges is of insufficient detail to provide accurate assessments of the amount of land that would be required for CDR, and the pledges themselves cannot be taken as precise descriptions of what will happen in the future, our analysis provides an initial estimate of the implications for global land pressure of national climate pledges. More transparency and consistency in country pledges would facilitate future analysis and is necessary to help answer one of the core CDR governance questions posed by the IPCC, namely: which CDR methods governments want to see deployed by whom, by when, at which volumes and in which ways[24].

## Results

We find that 990 (892–1087) million ha of land would be required to meet the aggregate CDR commitments in country climate pledges from 2020 to 2060. This area is in addition to land already counted towards climate targets, as we included only future land-based CDR in pledges. The aggregate land area is larger than the United States of America, at 983 million ha, or equivalent to two-thirds of the global cropland area, at 1561 million ha in 2020[25]. Of these, 435 (395–475) million ha would entail the conversion of existing land uses to forests or energy crops, while 555 (466–644) million ha would entail the restoration of degraded ecosystems, according to our categorisation of activities (Fig. 1).

### National climate pledges

National climate pledges express commitments in a range of different metrics and qualitative ambitions. Of the 194 countries reviewed, it was possible to quantify the land sector mitigation commitments for 140 countries (including EU Member States, which were assessed as a bloc). Pledged activities included land and forest restoration, tree-planting or reforestation, and (for just a few countries) BECCS or direct air capture and storage (DACS). We used three methods to estimate land area in climate mitigation pledges. When carbon removal commitments were not directly expressed as a land area (direct area pledges), we calculated the land area by converting pledges made in tonnes of $CO_2$ removed to land area via IPCC removal factors (emissions pledges). For pledges made as number of trees planted or percentage of forest cover expansion we used external data sources to estimate land area (indirect pledges) (see methods).

Twenty-four countries referred to land management activities resulting in carbon removal in their climate plans, but did not provide any information that was quantifiable. These include large forested countries

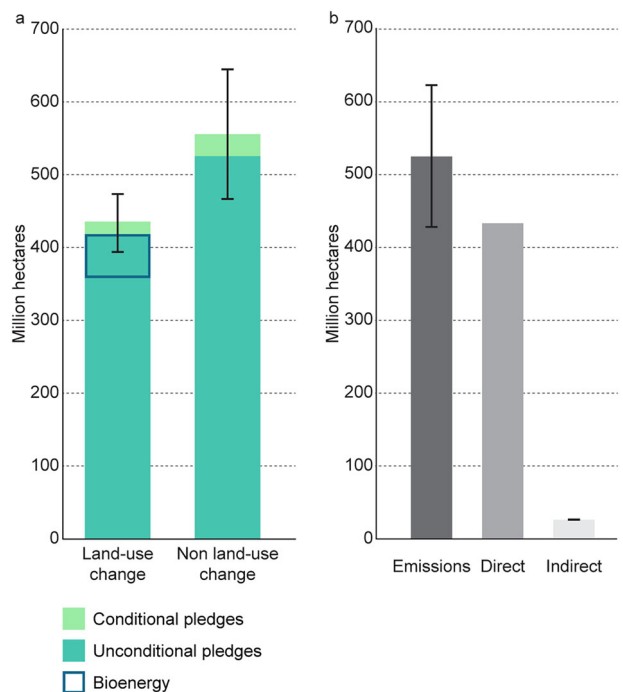

**Fig. 1 | Land use type and pledge type. a** Pledged CDR divided into activities likely to cause land-use change, and restoration of existing land uses (no land-use change). Land-use change occurs through conversion of non-forested lands to forest, or the conversion of forest or other natural lands to bioenergy crops. Restoration of existing land uses refers to restoring degraded forest, agriculture or wetland ecosystems. Unconditional pledges are shown in dark green while conditional pledges (additional actions pledged by developed countries that are contingent on climate finance) are shown in light green. The proportion of land-use change for bioenergy crops is shown in blue. **b** Pledged CDR divided into pledge type (direct land area, emissions based (tonnes $CO_2$ removed) or indirect area (pledge made as proportion of total forest, total land area or number trees planted)). Error bars indicate 2 s.d. on the mean.

such as Argentina, Central African Republic, Gabon and Peru. A further 27 countries made no reference to land activities for mitigation, including small island developing states such as Micronesia and Narau; arid countries such as Iraq and Kuwait; and very small states such as Monaco. Countries with large forest area that did not refer to land-based removals in their climate plans include Azerbaijan, Malaysia, the Philippines, and Serbia. In all, this amounts to 51 countries for which we could not quantify CDR (a further 3 countries have not submitted an NDC).

### Types of CDR in pledges

Country pledges contain a variety of descriptions of land management activities that we categorised according to IPCC land activity categories (see Table 1).

Table 2 provides an overview of how the land area is distributed across different types of CDR found in pledges. A key distinction is between those activities that propose reforestation or tree-planting that requires a change of land use, and those that restore existing land uses. This distinction is important because activities that involve changes to existing land uses are more likely to infringe on other societal concerns, such as food security and biodiversity protection.

Regrowth of managed forests, including replanting after harvest, does not constitute a land-use change according to IPCC guidelines[26], and so any reference to a changed management in existing forests was classified as anthropogenic restoration. Forest expansion, new forests or plantations were assumed to involve land-use change if existing forest was not mentioned. We have included energy crops in the land-use change category, although whether bioenergy crops will drive an

**Table 1 | Land activity categories and examples of interventions in national climate pledges**

| Activity category | Land management interventions | Examples from national climate pledges |
|---|---|---|
| Primary forest | Protecting existing intact forests | • By 2030, reduce deforestation to 80% compared to the baseline (Bolivia). |
| | | • Reduce the national deforestation rate by 50% by 2030 (Liberia). |
| | | • Maintenance of 100% of the native forest area (Uruguay). |
| Old secondary forest | Restoring existing degraded forest through changed forest management activities. | • Restoring 1.3 million ha for buffering and ecological restoration (Colombia). |
| | | • Assist natural regeneration of forests through different silvicultural methods on 7500 ha by 2030 in order to restore natural forest cover (Georgia). |
| Young secondary forest | Reforestation, forest expansion (mixed species) | • Enhance sinks and reservoirs of GHG through expansion of the forest cover by planting five million trees over the next five years (Sierra Leone). |
| | | • An additional carbon sink of 2.5 billion to 3 billion tonnes of carbon dioxide equivalent through forest and tree cover by the year 2030 (India). |
| Plantations | Commercial planting for harvest often monocultures | • Forest tree Plantations (2,000,000 trees) expected to reduce cumulative emissions by 247.36Gg $CO_2$eq (Jordan). |
| | | • Ten Billion Trees Tsunami Program (TBTTP) will sequester 148.76 MtCO2e emissions over the next 10 years (Pakistan). |
| Mangroves | Mangrove restoration or expansion | • Restore 4000 ha/y of mangrove (Senegal). |
| | | • Restore, enhance, and manage about 5000 ha of degraded mangrove resources over the next 10 years (Sierra Leone). |
| | | • Plant 30 million mangrove seedlings to enhance natural carbon sinks by 2030 (UAE). |
| Agroforestry | Trees in croplands, regenerative agriculture | • Plant 10,000 ha of trees per year under agroforestry: 0.358 MtCO$_2$e (Namibia). |
| | | • Several initiatives are aimed at combining agricultural and forestry activities to improve food security. Increase tree cover, both in urban and rural areas. 169 GgCO$_2$e in 2030 (The Gambia). |
| Silvopasture | Trees in grazing lands, restoration of rangelands | • Regeneration of 5.35 million ha for rangelands (Afghanistan). |
| | | • Incorporation of good management practices for the natural range and the breeding herd in 1,500,000 ha of natural pastures (Uruguay). |

**Table 2 | CDR Typology**

| Approach | | CDR type | IPCC activity type | Land area (ha) |
|---|---|---|---|---|
| Indirect | | Protection | Primary forest (not included as CDR) | |
| Direct | Conventional | Restoration 555 Mha (no land-use change) | Old secondary forest | 483,698,106 |
| | | | Mangroves | 407,809 |
| | | | Silvopasture | 48,539,027 |
| | | | Agroforestry | 22,514,974 |
| | | Reforestation 374 Mha (land-use change) | Young secondary forest | 345,613,136 |
| | | | Plantation | 28,115,569 |
| | Novel | BECCS 61 Mha (land-use change) | Bioenergy crops | 60,983,547 |
| | | DACS | No land area assumed | |

Approaches to CDR can be divided first into indirect (non-anthropogenic) and direct (anthropogenic) removals. Ongoing removals that are part of the land sink, such as primary forests, were not included in our results although they were sometimes included in country pledges[72]. Direct removals are often divided into nature and technology based. Both approaches were identified in country pledges. Conventional removals can be achieved through restoration (no land-use change) or reforestation (involving a land-use change). Novel removals identified in country pledges consisted of BECCS or DACS. We assumed no land area for DACS.

expansion of agricultural land is dependent on technological advancements such as yield increase, strong land governance and dietary changes[27,28]. Based on these classifications, more than 40% of the total pledged land area is devoted to reforestation, plantations or energy crops. The rest of the area is pledged for the restoration of degraded forests, agricultural lands, or coastal ecosystems.

Of the land-use change area, 61 million ha would be required for BECCS, with quantified values available for just five countries. While biomass feedstocks range from forestry and crop residues, to biogenic waste[29], to dedicated first and second generation energy crops[30], none of the countries pledging to use BECCS provide details on how the biomass would be sourced. Bioenergy capture rates vary based on choice of feedstock, yield rates and conversion efficiencies. In the absence of information provided by countries, we have assumed BECCS pledges are met via energy crops using country-specific yield values[30] and a 60% conversion efficiency rate following Vaughan et al.[31]. Different assumptions could reduce or increase the land area required

for BECCS, particularly technological advancements and dietary shifts[27].

We did not separately assess land area for bioenergy demand (i.e., bioenergy included in energy sector pledges without BECCS), as the focus is on CDR. We also did not quantify pledges for the protection of existing forests that would result in emission reductions. Neither did we include removals from primary forests (occasionally quantified in pledges) as these are non-anthropogenic removals included in the terrestrial land sink[32]. This is not to underplay the importance of maintaining intact ecosystems and their critical role in climate stabilisation[33], but reflects our objective to assess the land area required for additional carbon sequestration specifically.

Restoration commitments have elsewhere been quantified at 1 billion ha based on direct area pledges for 115 countries under the United Nations Convention to Combat Desertification (UNCCD), the Convention for Biodiversity (CBD), UNFCCC and Bonn Challenge[34]. These restoration pledges only partially overlap with the land area we have quantified in

climate pledges, which rely on both direct area pledges (often included in climate commitments and restoration pledges) and emissions-based pledges (only included in climate commitments). Not all restoration pledges made by countries have been included in the climate mitigation pledges we assessed. If climate and restoration pledges were taken together, the area would likely be larger than our total of 1 billion hectares. This speaks to the uncertainty of presenting land pledges without spatial analysis, and to the lack of coordination between different environmental conventions, which exacerbates the difficulty of assessing trade-offs and conflicts between different societal objectives.

### Temporal distribution of land in climate pledges

Looking at the temporal distribution of pledges shows that countries rely on an additional 211 million ha of land for carbon removals by 2030, which scales up to a total of 990 million hectares by 2060 (Fig. 2). Net-zero pledges for 2050 and 2060 include larger areas of land, meaning the reliance on land in mitigation pledges can be expected to increase as more countries make net-zero pledges. This resonates with recent analysis which shows that CDR is expected to play a larger role in mitigation pledges towards and beyond 2050[4].

Our results indicate a remarkable rate of land use change of up to 13 million ha per annum if the reforestation component of pledges is assumed to scale up linearly from 2020 until 2050 (with only 50 m ha of reforestation to take place after 2050). Comparing these country intentions to past peaks in land appropriation may be indicative of the risks involved. For example, over the period 2007–14, which was the most intensive period of what has been dubbed 'the global land rush', an average of seven million ha was transacted per year in the Global South. This development was seen as a great threat to small scale farmers' land tenure security and livelihoods[35]. Monitoring of large-scale land transactions shows that by 2020 between 30 and 73% of this area has been converted to agricultural production, with far-reaching consequences for rural livelihoods and natural habitats[36].

Future scenarios provide another point of comparison. Modelled pathways that limit warming to 1.5 °C with no or limited overshoot show increases in forest cover for carbon removal of 322 million ha (median, with a range of −67 to 890 million ha)[16]. Many of these

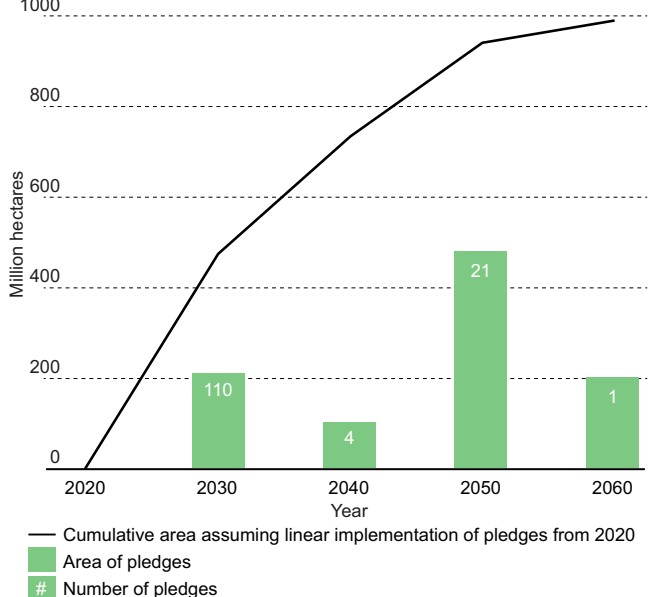

**Fig. 2 | Carbon dioxide removal in climate pledges over time.** Total land area for climate mitigation will increase over time as pledges are implemented (cumulative line depicts linear scale up of pledges from 2020), with 110 pledges made for 2030, 4 pledges for 2040, 21 pledges for 2050, and 1 pledge for 2060.

pathways also include large amounts of energy cropland area, to supply biomass for bioenergy and BECCS, with 199 (median, 56–482 range) million ha by 2050[16]. While these scenario ranges do not reflect a statistical sample[37], it is illustrative that delivering the median scale of CDR projected in these pathways would require an average land-use change of 17 million ha per year between 2020 and 2050.

Hence, our analysis suggests that the rate of direct land use change for carbon removal included in national climate pledges, at 13 million ha per year, is unprecedented from an historical perspective. Furthermore, it is comparable to the average rates of land transformation assumed in global modelled scenarios by mid-century that have raised concerns within the scientific community exactly over their vast consequences for land use, governance, and rural livelihoods[38,39].

### Geographical distribution of land in climate pledges

A few countries have pledged a large proportion of the aggregate land area. Just four countries (Russia, Saudi Arabia, the US, and Canada) contribute over 70% of the global total and the ten largest countries amount to 85% (Fig. 3). These are all large countries and all major fossil fuel producers. Of the 10 largest country pledges, only Saudi Arabia and Australia explicitly mention that they intend to use internationally traded credits or tree planting in other countries to help meet their pledges.

Russia has the largest CDR area, based on a pledge to more than double the absorptive capacity of managed ecosystems[40], which we calculate as 350 million ha of forest regrowth or 21% of Russia's total land area. Around ten percent of this could potentially be achieved by regrowing abandoned crop land[41]. As part of its plan to reach net zero by 2060, Saudi Arabia has pledged to plant approximately 40 billion trees in neighbouring countries, equivalent to 200 million ha[42]. This is part of the Saudi hosted Middle East Green Initiative, which includes a strong focus on tree-planting[43]. The large land area pledge of the US is explained by a modelled reliance on reforestation and technological CDR of 1000–1800 Mt $CO_2$ per year by 2050[44]. The strategy is not spatially explicit, but we estimate 54 million ha would be required for reforestation and a further 54 million ha for BECCS to deliver the upper end of expected CDR.

Other countries with the 10 largest land area pledges for CDR are shown in Fig. 3. India has promised to create an additional carbon sink of 2.5–3 billion tonnes of $CO_2$ through reforestation by the year 2030, requiring additional forest area of 24 million ha, or 8% of India's land area. India's national policies to increase tree cover have already been criticised for failing to consider questions of land ownership, existing land use patterns, and ecological factors such as suitable tree species and elevation[45]. Australia's land area pledge is inflated by a large reliance on BECCS to meet its 2050 targets, despite no policy discussion to date on this approach, and includes internationally traded forest carbon credits[46]. China pledges to increase forest coverage to 25% by 2030, requiring an expansion in forest cover of some 19 million ha from 2020[47]. This is part of a decades-long effort from China to combat land degradation and desertification, and now climate change, with tree-planting[48,49]. On the restoration side, we counted a large area of forest restoration in the EU, based on the 2023 LULUCF Regulation[50], which for the first time includes biological land removals in the EU climate target. Brazil and Ethiopia have pledged large areas of land for restoring degraded forests and pasturelands, with smaller areas of forest expansion.

When calculating the share of national land area devoted to CDR, many low-income nations stand out, particularly in Africa. Figure 4 shows the 16 countries with the largest share of their land area pledged for CDR. Only five of these countries do not fall into the category of least developed countries (LDCs) or most vulnerable countries (including Africa). This could substantiate fears of an impending land rush[36] and resulting dispossession of rural communities as these poorer countries plan to devote substantial parts of their national land area to CDR efforts. Conversely, given that more than half of emissions from many low income countries originate from land-use and are bound to development and

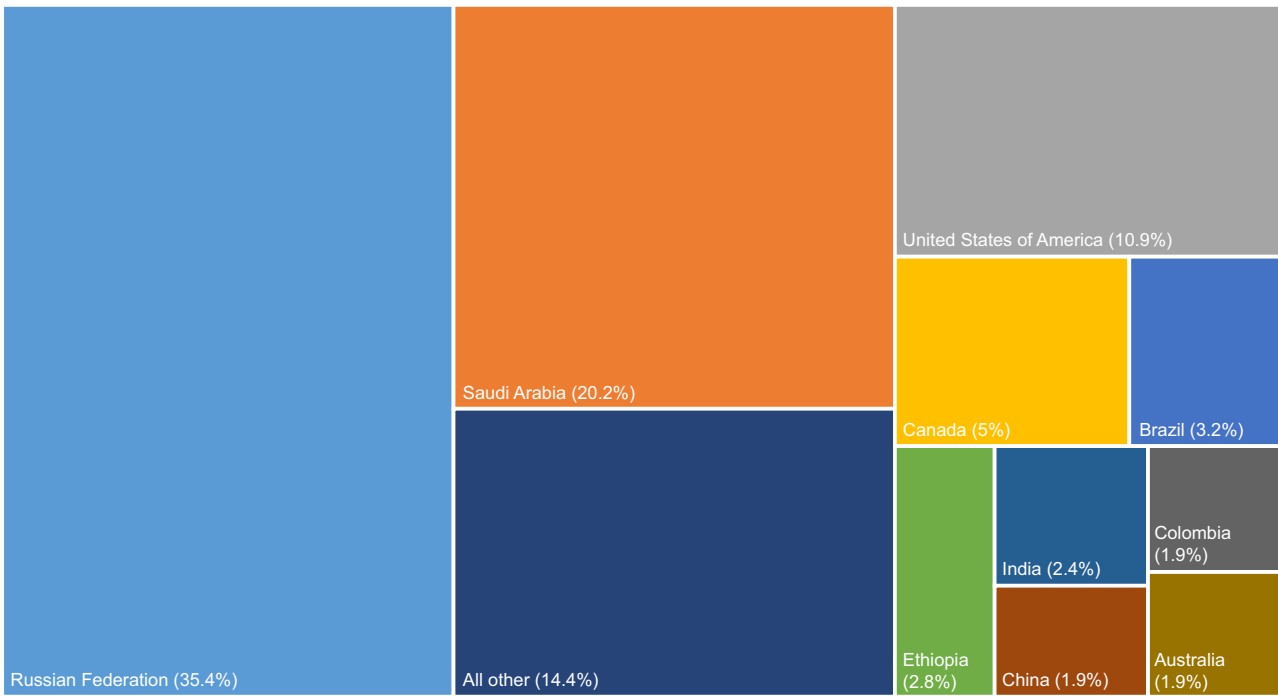

**Fig. 3 | Distribution of land area for CDR pledges among countries.** The countries with the 10 largest land area pledges are indicated by relative square sizes represent relative proportion of total pledged land area for each. Colours are used only to differentiate cells.

food security imperatives, we can expect a greater focus on the land sector in these countries' climate plans. Most of the pledges from LDCs focus on agricultural regeneration rather than reforestation or tree-planting - activities that can take place over large areas in agriculturally dependent countries without displacing people or food production. The two countries that have pledged more than 100% of their land area in NDC targets (Dominica and Equatorial Guinea) are explained by a combination of direct area and emissions-based pledges being made, where the governments may not have considered, or may use different methods to arrive at, the land area required for emissions-based CDR pledges.

## Discussion

Our findings point to enormous expectations for land to meet climate mitigation goals under the Paris Agreement. The scale of land-based removals in national climate pledges speaks to the risks created by net zero targets that are over-reliant on land-based CDR. Such risks include that expected future removals undermine near-term emissions reductions, or that non-permanent removals with a high risk of reversibility will be used to compensate for essentially permanent fossil fuel emissions[51]. Our results should be considered a conservative estimate of countries' intended reliance on land for mitigation due to several limitations of this study. First, there are 51 countries for which we have no estimate, including several larger forested countries. Second, longer-term pledges indicate a trend for even larger reliance on land and most countries have not yet submitted 2050 climate pledges, indicating that land area in pledges will increase as more countries make net zero commitments. On the other hand, net zero targets suffer from vagueness, meaning that many assumptions must be made to estimate demands on land implied in country pledges. For example, the amount of land potentially required for BECCS deployment is highly uncertain, given the wide range of bioenergy capture rates (depending on feedstock, yields, conversion efficiency), yet pledges do not specify any details to indicate land area requirements. BECCS pledges could be met at least partially through biogenic wastes, yet the scale of pledges so far from just two European countries would require 50% of Europe's potential waste supply[26]. Given that the largest land

areas were calculated from emissions pledges rather than direct area, our findings may instead overestimate the amount of land governments intend to use for mitigation.

Limitations notwithstanding, our results are concerning from the perspective of climate mitigation. For one, the decade-long periods often required for biomass to reach full carbon removal potential mean that CDR pledges may not deliver their expected removals within the pledged timeframe. Indeed, analysis shows that 2050 pledges put us on track for 1.9 °C compared to 2030 pledges only[52], yet we find the majority of reliance on land is in 2050 pledges, potentially undermining expected ambition in longer-term targets. This indicates a need for more clarity and transparency across governments' climate and land restoration pledges, and a separation of emission reduction and removals as well as land sector and energy targets to simplify reporting and avoid undermining mitigation goals[20,53].

Another risk highlighted by our results is the extensive reliance on reforestation in climate mitigation pledges. Tree-planting approaches have gained prominence as a mitigation activity with estimates for large scale CDR[54,55]. Yet establishing new plantations or expanding forest areas requires a land use change, which is also the leading driver of global biodiversity loss[8]. Reforestation and tree planting efforts risk increasing competition over land and could have negative repercussions on biodiverse ecosystems including grasslands and existing forests, food sovereignty, and vulnerable and land-dependent peoples' tenure and livelihoods[56]. Restoration approaches could also lead to displacement and dispossession due to stricter regulations on resource access and uses and enforcement of these[57,58]. The way these policies play out and how land is used will depend upon local land tenure as well as other social and economic factors. Biophysical effects also impact whether increased forest cover leads to increased cooling or warming - in the tropics forest cover can improve local climate conditions, such as through cooling and increased rainfall[59,60].

It is alarming that the extent of land required for CDR in government climate pledges already tracks against the upper end of mid-century scenario expectations for reforestation, with only 5 pledges made for BECCS to date. There are well founded concerns that land use

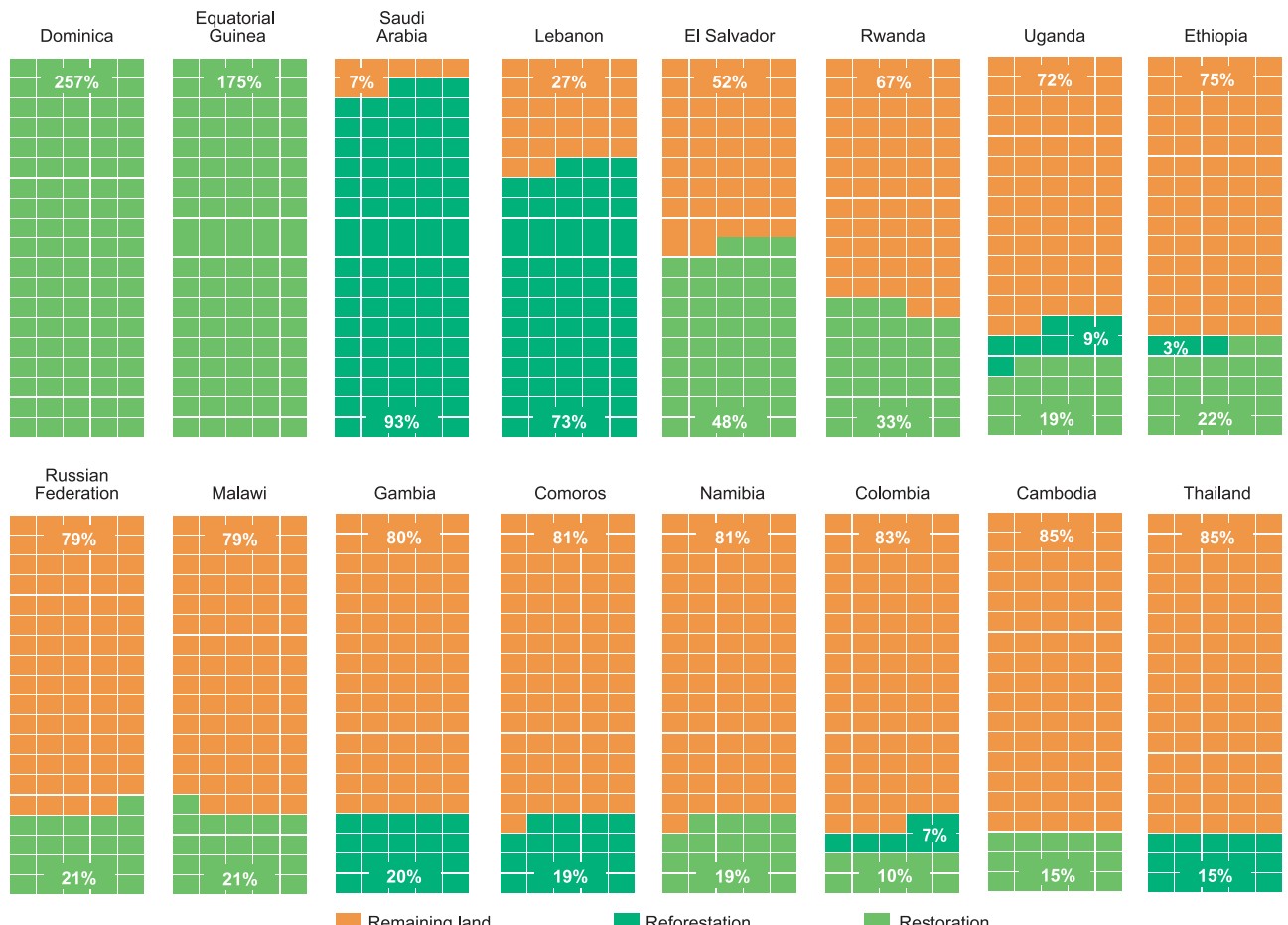

**Fig. 4 | The 16 countries with the largest share of national land area pledged for CDR.** Total national land area represented by each box, with light green showing the proportion of total land area pledged for restoration activities (non land-use change), dark green shows proportion of land area pledged for reforestation and energy crops (land-use change) and orange shows proportion of remaining land.

change on this scale would be particularly pronounced in the Global South, where historical trends of pasture and cropland expansion would need to be reversed, leading to an absolute reduction in these land uses[35]. This would occur at the same time as populations in these areas are expected to grow and climate impacts over the next decades will mount, increasing vulnerability and hitting food production with more unpredictable and extreme weather[61,62].

Our results show that many countries are planning large-scale land-based CDR to meet their climate targets, including an aggregate area of afforestation and tree planting that would amount to almost one-third of current permanent global cropland area. Given that mitigation scenarios to limit warming to 1.5 °C also require CDR to reach net-negative emissions in the second half of the century, the pressure on land is only likely to increase beyond 2050. Hence, these results which are based largely on 2030 and 2050 pledges for a subset of the world's countries, already show a worrying trend for reliance on land in national climate mitigation strategies.

It is well known that current national climate pledges are not on track to limit warming to 1.5 °C[34]. Our results show that these pledges embed another layer of insufficiency in that they embed unrealistic expectations of the land sector. The geographical distribution of land claims in our dataset illustrates that especially major fossil producers with large land areas are betting on land-based CDR. This pattern could suggest that countries push the mitigation burden onto the land sector rather than phasing out emissions from fossil fuels and land-use change. We argue that the scale of CDR already included in real world pledges implies direct land-use changes at an unprecedented rate that

could jeopardise food security and biodiversity goals. Far from calling for scaling up CDR to close the gap, we therefore call for greater transparency around the approach to land management in climate mitigation plans, prioritising restoration of degraded lands over tree planting and forest expansion, and scaling up ambition levels in near-term emissions reductions rather than striving to achieve what appear to be unrealistic targets for land-based CDR.

With the second round of NDCs due in 2025, countries need to be more ambitious in the emission reductions they deliver for 2030 and beyond to limit reliance on CDR. We urge governments to be more explicit about the land area required in land-based climate mitigation commitments, both to facilitate national discussions and planning around land use, and to ensure CDR components of mitigation pledges are not overstated.

## Methods
### Sample
Climate pledges were reviewed for 194 countries. The European Union and its 27 Member States communicated one joint NDC, hence we have analysed the climate pledges of the EU as a bloc, rather than individual Member States. Three countries have not submitted an NDC, meaning that pledges for 164 countries plus the EU were assessed. To assess the reliance on land in these national pledges, we reviewed all LT-LEDS or NDCs submitted to the UNFCCC up until the end of 2023 (with the exception of Brazil for which we assessed the 2015 NDC following government announcements to reinstate that NDC[63]). We focused on LT-LEDS as a proxy for net zero and 2050 targets. That is, we assessed

the longest-term pledge that was available, assuming that any land-based CDR in near-term pledges is encompassed in longer-term pledges. For countries without long-term pledges, we reviewed NDCs. For a handful of countries that had not included CDR in their UN submitted climate pledges we included other government statements.

## CDR typology

Country climate strategies and pledges express commitments in a range of different metrics and qualitative ambitions. Therefore, a number of assumptions were made to identify the scale of CDR commitments, and the associated land area required. CDR was presented in climate pledges as a range of different land management activities, or technology or emission based commitments, including: emissions reductions required to achieve net zero or interim (2030) targets compared with total emissions (presented in Mt $CO_2$e or percent of total emissions); references to residual or remaining emissions at the time of net zero; reference to removals/sequestration/CDR (presented in Mt $CO_2$e or proportion of total emissions); direct references to land area (in hectares, acres or $km^2$) or proportion of land area (of country, or of a land cover type, i.e.,: proportion of forest cover to be maintained or extended, and as number of trees to be planted). We categorised these pledges into three types: direct land-area pledge, indirect land-area, or emissions-based. Direct area is reported directly from the pledge, while indirect and emissions-based pledges are converted into land area through the methods described below.

The various approaches to land management activity types in national climate strategies were categorised into seven activity types, based on their carbon sequestration potential (using IPCC removal factors). Table 1 shows the seven land-use categories used, in relation to ecosystem condition, with country examples that highlight the key search terms used. 'Primary forests' are intact natural forests with minimal disturbance (key terms – primary, protected). 'Old secondary forests' were selected to represent regeneration of degraded natural forests (key terms - restoration, forest management, forestry).'Young secondary forests' refers to establishing new forests of mixed species (key terms - reforestation, forest expansion). 'Plantations' was used when countries referred to establishing commercial forests or monoculture forests (key terms – afforestation, plantations) 'Bioenergy' was used when countries referred to BECCS deployment (key terms - BECCS). Agricultural landscapes were classified into two broad categories – 'Agroforestry', for pledges that referred to agricultural regeneration or integrating trees into agricultural landscapes (key terms – regeneration, agroforestry, mixed uses), and 'Silvopasture', for pledges that referred to restoring degraded rangelands (key terms – rangelands, silvopasture). The activity type 'Mangroves' was used to quantify the removal potential of restoring or expanding mangroves. This categorisation represents a simplification of the range of land management activities and practices that countries have referenced in their climate strategies, and which result in CDR.

Table 2 characterises these seven land management categories based on whether the primary intervention involves protection, restoration or replanting, and shows the land area for each activity type. Pledges for avoided emissions and the protection of existing forests were noted, but not quantified in the context of our aim to assess the land area required for carbon dioxide removal in national climate pledges. For agricultural activities, removal factors were sourced from the IPCC[64]. For forestry activities, removal factors were sourced from ref. 65 For bioenergy, country specific median yield values were sourced from ref. 30, and a conversion efficiency of 60% was assumed following ref. 31. Subjective choices around conversion efficiency, which varies from 40% to 90% across studies, can impact the results in terms of land area required, and hence the perceived feasibility space (See Table S1 for details of calculations used). It should be noted that climate effects on energy crop yields are not captured by Li

et al.[30], and have already been observed to reduce agricultural productivity[62]. Similarly, climate change effects, such as $CO_2$ fertilisation on forests are not captured by Harris et al[65]. With future changes in climate, carbon storage capacity may change over time, increasing through factors like $CO_2$ fertilisation, which may weaken over the century as climate effects start to take over from $CO_2$ fertilisation and other effects[66]. These indirect, climate and $CO_2$ feedbacks are uncertain and remain model-dependent[67–69].

## Converting emissions based and indirect pledges to land area

For emissions-based pledges, default removal factors from the IPCC were applied based on the activity type and climate domain of the country (or implementation area, if this was identified as being outside the pledging country) for the CDR typology derived as described above. A more accurate representation of the variety of land management activities would entail considerably more work, but would not greatly change the results, given that the range of emissions removal factors that can be applied is limited. A sensitivity analysis using a global average removal factor resulted in an 8.4% increase in land area (117 million ha) showing that the use of biome and activity specific removal factors constrains the calculation of land area, but selection of activity types does not determine the results. Removal factors are based on aboveground biomass only. Including a belowground biomass increment for relevant activity types decreases total land area by 8.9 million ha, which is less than 1% of the total land area (See Supplementary Data[70] and Table S3).

For indirect pledges made as a proportion of forest cover or land area, we used data from publicly available datasets on land cover and land use from the Food and Agriculture Organization of the United Nations (FAO), and national GHG emissions profiles such as the Climate Analysis Indicators Tool, to calculate the implied land area when not directly stated. For indirect pledges made as number of trees planted, tree density per hectare was taken from Crowther et al.[71], depending on the ecoregion of the country in question (See Table S2).

## Uncertainty analysis

We estimated uncertainty in the land area estimates from emissions-based pledges and indirect area pledges at the scale of climate domain and activity type because the estimates were computed from uncertain values. We followed a propagation of uncertainty approach, which reflects calculations of variances (and hence, s.d.). We note that the uncertainty analysis, being based only on the denominator to calculate land area, will be an underestimate. Many other variables contribute to uncertainty in the aggregated data, including biome classifications and assumptions made in interpreting country statements.

For emissions-based pledges, uncertainty estimates were based on uncertainties in the removal factors from ref. 65 for forest land, IPCC[64] for agricultural land, and ref. 30 for bioenergy. For indirect pledges, uncertainty estimates were based either on tree-density s.d. from ref. 71 or where total forest or land area reported to the FAO[25] was used, no uncertainty estimates are available. For direct area pledges, we assumed the uncertainty to be zero. Table S1 shows the source and contribution of the relevant denominator, along with its associated uncertainty. Mean and s.d. values shown in figures are based on an aggregate of the relevant data rows in the Supplementary Dataset[70].

The land required s.d. was computed using a standard formula depending on which method was used to compute the land required, assuming that the mean of the tree density or removal factor is sufficiently larger than zero.

$$\text{land required} = \frac{\text{tree density SD}}{\text{tree density}}$$

or

$$\text{land required} = \frac{\text{removal factor SD}}{\text{removal factor}}$$

## Data availability

The datasets generated in this study have been deposited in the fig-share repository, available at https://doi.org/10.6084/m9.figshare.24080472. Summaries of key data and extended methods are provided in the Supplementary Information and the Source Data file. Source data are provided with this paper.

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

## Acknowledgements

The authors would like to acknowledge the work of Nathan Ivetic, Yann Robiou Du Pont, Heather Keith and Muhammad Luqman on data input to the Land Gap Calculator, as well as the assistance of Emma Johansson in rendering figures. K.D., K.L.C., J.F.L., and W.C. were supported by the Independent Research Fund Denmark (0217-00078B). W.C. was supported by the Swedish Research Council for Sustainable Development (FORMAS), dnr. 2019-01953. K.D. was supported by the Australian Research Council, DE230101175.

## Author contributions

K.D., K.L.C., J.F.L., and W.C. conceived the idea for the study and wrote the main paper. K.D. and A.S. led the data analysis, supported by all other authors. All authors discussed the results and implications and commented on the manuscript at all stages.

## Competing interests

The authors declare no competing interests.
