## [Transparent Peer Review file · Nature Communications]

Over-reliance on land for carbon dioxide removal in net-zero climate pledges

Corresponding Author: Ms Kirstine Christiansen

Version 0:

Reviewer comments:

Reviewer #1

(Remarks to the Author)

Review of "Over-reliance on land for carbon dioxide removal in national climate pledges" by Kate Dooley, Kirstine Lund Christiansen, Jens Friis Lund, Wim Carton
Nature Communications

Recommendation: Revise and resubmit

Key results

The authors reviewed country pledges that express climate commitments related to land and that are represented as a range of different metrics and qualitative ambitions. Several countries (53 of ~165, or ~32%) did not provide enough information for the authors to include in their assessment. This compilation exercise by the authors is in itself commendable, as the information provided by countries tends to lack transparency, making it difficult to synthesize the information as was done in this paper. The authors' recommendation in the conclusion is well supported that greater transparency is needed around the approach to land management in climate mitigation plans and the assumptions made more clear about the land area needed for their land-based mitigation commitments.

From their review, the authors conclude that based on current pledges, approximately 1.1 billion ha of land would be needed globally for land-based CO₂ removals to be delivered as pledged by countries over the time period 2020-2060, an area equivalent to two-thirds of global cropland area. From this analysis, the authors suggest that these commitments place too much expectation on land to deliver on the Paris goal of achieving a net balance between anthropogenic emissions by sources and removals by sinks of GHGs in the second half of the century, potentially undermining the need for near-term emission reductions.

Validity and robustness of conclusions

- The type of analysis as conducted in this paper is helpful, but mainly as an example of how the analysis the authors WANT to conduct in a credible manner is simply not possible based on the information provided. I would like to see it re-written with this angle, rather than trying to pass off the analysis that was done as something analytically robust. As currently written, it is more appropriate for a specialized policy audience as a back-of-the-envelope estimate for those following UNFCCC negotiations. As currently formulated, the estimates lack the analytical rigor I would expect from a high-profile Nature publication and the findings do not represent a particularly significant or novel advance to specialists in the field. This is less the fault of the authors and more the fault of the bad data they had to work with.
- The main push of the paper is to voice concern over the amount of land required to remove carbon dioxide at the scale and pace countries have pledged as a way to get us out of this global climate mess. To put this in context, it could be helpful if the authors mention country pledges on the emission reductions side as well, not just for land but overall across sectors; are these equally concerning or is the concern limited just to CDR on land? Are emission reduction pledges more realistic than the land CDR pledges or similarly unrealistic? Might be worth making a point up front that these commitments/pledges may be set by countries to be overly ambitious on purpose, to be used more as aspirational targets/tools for political posturing than as having any basis in reality. The example of corporate commitments made in 2014ish to end deforestation by 2020 comes to mind: was that goal achieved? No. Was it a useful pledge to rally around a collective vision of hope? Yes. Once that narrative is set up about what role these pledges play, and how the Paris Agreement encourages ratcheting up of ambition over time, then the authors could make the recommendation/conclusion backed by their analysis that pledges should be more realistic and transparent about the additional climate mitigation that CDR from the land sector could actually

provide.

- The authors acknowledge that their analysis required many assumptions to arrive at their estimates. Based on my review of the online methods and the excel spreadsheet, these assumptions are not as clearly outlined as I hoped they would be for a reviewer with limited time. More straightforward explanation would be helpful about how the land area estimates per country were calculated and more explanation provided in what assumptions were used where. The authors should clearly acknowledge the limitations of their analysis and that the result from their uncertainty analysis is surely an underestimate.

Data and methodology

- The authors' division of land into reforestation involving a change in land use vs. restoration of degraded forests is somewhat puzzling to me. The authors imply that reforestation involves a land-use change, but depending on country context reforestation may be synonymous with restoration of forest land remaining forest land. For example, are trees growing back after harvest – as may be prevalent in three of the four countries identified that contribute substantially to the global total land area for CDR (Russia, US and Canada) - counted by these countries as reforestation or restoration of a degraded forest? Regrowth of young secondary forests and plantations may not result in a change of land use because it may be considered by countries as forest land remaining forest land.
- It was difficult for me to follow the specifics of what was happening in the cells of the Land Gap Calculator spreadsheet. It would be helpful to include more information in the Notes tab, and/or provide further explanation in the supplementary Word doc, of a worked example of how estimates are derived for different scenarios when carbon removal commitments were not expressed directly as land areas. For example for each of the pledge types, provide an illustrative example of how the authors translated it into an estimate of land area. Saudi Arabia was based on a pledge on number of trees – the conclusions of the paper depend heavily on how that number of trees estimate was translated into an area of land and what assumptions were used. Currently all that is hiding in spreadsheets and supplementary material, but it's critical to elevate the assumptions used to the main text if the paper's conclusions are to be supported.
- Similarly, Russia's importance to the overall geographic distribution of land in climate pledges (33% of the total) would be particularly important to understand how the estimate was derived, since it was not a direct pledge of land area. How does a pledge to "more than double the absorptive capacity of managed ecosystems" translate into the assumptions made by the authors to arrive at the land area needed? These country pledges may be designed to be vague on purpose, and that point could be made in the paper.

Analytical approach

- I suggest the authors review national GHG inventories to understand what level of CO₂ removals are reported by countries, to put the CDR pledges into context.
- It is not clear to me how the mean and standard deviation of areas of restoration and reforestation were calculated in Figure 1.
- The authors didn't assess bioenergy demand or quantify pledges for protection of existing forests that would result in emission reductions, or include CO₂ removals from primary forests as these are non-anthropogenic removals included in the terrestrial land sink. Focus is on additional C sequestration specifically. More clarification should be provided on how countries account for current CO₂ removals in secondary forest and plantations, vs. additional CO₂ removals from "restoration" or "reforestation" of secondary forest. See also Nabuurs et al. 2023 (Communications Earth and Environment) who argue that carbon dioxide fluxes from all forest land (managed and unmanaged) need to be recorded by countries in order to help track progress towards global climate targets.
- Uncertainty analysis: Note that removal factors and their uncertainties from IPCC 2019 refinement for temperate forests were revised in a correction published by IPCC in July 2023.
- This may be beyond the scope of the paper, but some reference could be made to the biophysical processes that may enhance or diminish the climate effects of carbon released or absorbed from forest biomass (e.g. albedo), particularly if a substantial portion of land-based CDR is expected to come from temperate and boreal regions (Russia, Canada, USA).

Suggested improvements

- Use active voice (We provide a first estimate" vs. "this paper provides a first estimate")
- Per Nature guidelines (I think), avoid claims of novelty ("it provides the first assessment of its kind")
- Abstract and throughout: avoid vague statements like "implying impacts on people and food security". What kind of impacts?
- Include in the introduction more context for the general reader of what the Paris Agreement goal is – achieve a net balance between anthropogenic emissions by sources and removals by sinks in the second half of the century. Important to include anthropogenic and to stress the role of forests on the sinks side.
- Perhaps worth pointing out that the idea that all countries achieving net-zero within their own boundaries doesn't necessarily make sense because all countries have a different starting point when it comes to carbon dioxide removals on land. Some developing countries are already net zero/net sinks! So it's an important point to make that to increase CO₂ removals beyond those that exist in the country now, we need to assess the level of ambition in climate pledges and what CDR would be on top of current CDR in the land sector.
- Another point to highlight might be how land use history is likely to play a role in how CDR actually plays out – how much CDR an ecosystem can support may differ from the CDR that countries are pledging to deliver. The way that these policies play out and how land is actually used will depend upon local land tenure as well as other social and economic factors.

Clarity/context

- "We present a breakdown of what these removals would look like" – this is too vague, suggest deleting from abstract. More clear: "We present a breakdown of how demands for land would be distributed geographically and over time"
- "For more than half of this area, the pledges envisage the conversion of existing land-uses to forests, while the remaining area is for restoration of degraded ecosystems." What is a "degraded ecosystem" as defined by various countries? How

much overlap is there between existing land uses (several of which are degraded from their natural state) and degraded ecosystems? Not clear how this breakdown was determined after reading through the methods.

- Not clear how land for reforestation/plantations for BECCS is calculated
- Introduction: "many climate mitigation approaches that rely on land, such as large-scale afforestation, threaten to exacerbate rather than address the biodiversity crisis." Change to "some" approaches, as many climate mitigation approaches that rely on land have positive biodiversity benefits if implemented well.

Reviewer #2

(Remarks to the Author)

The paper, entitled "Over-reliance on Land for Carbon Dioxide Removal in National Climate Pledges", makes a significant and relevant contribution to the climate policy literature, focusing on the assessment of land requirements for land-based mitigation options and the associated risks. As a publication based on the analysis of the "Land Gap Report", it provides a timely contribution to the scientific literature, esp. in the context of next NDCs due in 2025. However, there are several issues that require attention and major revisions before publication.

1. Relevance of the documents for domestic climate policy making:

- a. There is a flurry of analyses of NDCs and LT-LEDS. What is usually missing is a reflection on the political role of these documents. The lack of submitted LT-LEDS, for example, indicates the low political priority that UNFCCC signatories attach to these reporting mechanisms. It should be emphasised that not only is the implementation gap between actual policy and NDC/LT-LEDS huge, but also that these strategy documents have limited relevance for national climate policies and that these political contexts for specific pledges cannot be identified from these documents. This is not to say that a quantitative analysis of the documents should not be done, but the framing of such an analysis should take into account this limitation arising from the data source of strategy documents that may not fully reflect a country's policy priorities.
- b. The paper should provide additional interpretation and policy contextualisation, particularly regarding the high numbers from individual countries, such as Russia. This will help to interpret the high headline number reported at the beginning of the paper.

2. Methodology:

- a. The methodology used to extract data from the documents should be clarified. At present, the supplementary material lists the sources of the documents and calculations but not the exact wording on which the calculations are based. This makes it difficult to trace the commitments and check for consistency and accuracy.
- b. The paper currently categorises all types of land restoration as mitigation/CDR commitments. Given that land has been managed for a long time and that objectives have changed (e.g. combating desertification, biodiversity,...), it might be worth reflecting that CDR may not be the primary objective of all the pledges collected here, but could also be a side-effect/co-benefit of an initiative that policy makers had in mind for other reasons - it may not be all about CDR. It would be useful to distinguish between pledges that are explicitly focused on carbon removal and those where CDR is a secondary or co-benefit. This clarification will provide a better understanding of the nature and intent of the pledges.
- c. Related to Issue 1: The use of government documents (instead of NDC/LT-LEDS in the analysis should be explained (why are government documents used in some cases?) There are significant gaps between NDCs/LT-LEDS and actual national policy making (see also Issue 1). This should be better explained - or focus on only one type of document. Alternatively, the paper could consider focusing on one type of document to provide a more consistent analysis.

3. Permanence/reversibility of CDRs:

- a) The issue of reversibility and different permanence periods for different types of sequestration is critical, but is not addressed in the pledges and is not explicitly raised in the paper. Given the aim of the paper to inform the review of NDCs, it is important to highlight the challenges associated with relying on LULUCF-based CDR to meet climate goals. The potential measures to incorporate them into CDR policy, such as buffer pools and equivalence discounting, should be raised to highlight that land requirements could be even higher if LULUCF-based CDR is responsibly governed to meet climate goals.

In conclusion, the paper makes an important contribution to climate policy by addressing the role of land-based mitigation options in national climate commitments. In order to improve the relevance of the paper, it is recommended that the above issues are addressed in a major revision. By doing so, the paper can provide a more comprehensive and insightful analysis of the issue and have a greater impact on the climate policy discourse.

Reviewer #3

(Remarks to the Author)

I noted reading this paper and consider it a very worthwhile contribution.

The paper evaluates the scale and potential land-use conflicts (and other challenges) for nations to deliver on pledges around LULUCF and BECCS. I address and highlights the uncertainty and risk of presenting these land pledges in the absence of robust local assessment and especially spatial analysis. The same risk applies to national pledges as they pertain to energy efficiency, renewable energy expectations, and CCUS (especially storage). This lack of spatial granularity obscures a multitude of risks and uncertainties when it comes to resource capacities, environmental considerations and local values.

In the case of the land sector, the lack of coordination between different environmental conventions is especially problematic.

The paper is timely, as we approach COP28, proposals to update NDCS, and the IAMs modelling updates ahead of the 7th Assessment Report.

Acknowledging that I am not an expert in land sector analysis, the analytical work done appears robust and uses sound sources. The authors also acknowledge clearly the uncertainties and limitations on the analysis but I agree they have landed at a conservative place.

The paper is also very well written. I had a couple of very minor comments:

On page 6, where the authors write: "For example, the global land rush of the 2000s, which was seen as a great threat to small scale farmers, saw no more than seven million ha being transacted per year." Authors could perhaps add some descriptive context noting the broad readership of Nature Comms. E.g.:

"For example, the global land rush of the 2000s for the purpose of industrial-scale agriculture and resource extraction..., which was seen as a great threat to small scale farmers, saw no more..."

Also suggest the insertion of "approximately" or "around" in a number of places, given the acknowledged uncertainty in parameter. e.g. "Saudi Arabia has pledged to plant an additional 40 billion trees in neighbouring countries, equivalent to (around) 200 million ha".

Otherwise I recommend for publication.

Version 1:

Reviewer comments:

Reviewer #2

(Remarks to the Author)

Dear authors,

thanks for the careful revision of paper and considering the points raised by the reviewers. The improved transparency in the methods and the new framing about lack of data/details in the pledges improve the paper.

Two minor things I'd like to raise:

- the direct quote from the PA is not the exact quote (at least not from Art. 4
- I think in the supplementary material are a few notes (eg. U6, Z29) and colour (AF17) that should be deleted before publication

Kind regards

Reviewer #4

(Remarks to the Author)

The paper evaluates the reliance on land for CDR in national climate pledges. I generally find this topic to be highly important and the results noteworthy. The article is likely to have an impact on the field. I found results largely supported by data, although I think some revisions are needed. I provide some comments below. Thank you for the interesting read.

1. BECCS pledges seem unlikely to be forest feedstocks. In general, biomass residues are very important in ramping-up biomass supply at low levels of demand, whilst dedicated bioenergy crops dominate when demand exceeds 100 EJ year⁻¹ (Hanssen et al., 2020). The energy yields of short-rotation bioenergy crops exceed managed forests. Lignocellulosic bioenergy crops are considered in nearly every IAM, and managed forestry for bioenergy is often excluded due to sustainability issues (Daioglou et al., 2020). Sugarcane cultivation is a key contributor to modern bioenergy supply (Ramirez Camargo et al.). It cannot be assumed that BECCS biomass supply will be coming from forest plantations or claim that this is the most conservative land use estimate, please revise text and calculations. There is a need to go down to country-level detail and identify the most likely feedstock, and maybe also do some different scenarios with varying feedstocks (residues, energy crops, etc.). I think data from Li et al. (2018) or Li et al. (2020) could be useful.

Daioglou, V., Rose, S.K., Bauer, N. et al. Bioenergy technologies in long-run climate change mitigation: results from the EMF-33 study. *Climatic Change* 163, 1603–1620 (2020). <https://doi.org/10.1007/s10584-020-02799-y>

Hanssen, S.V., Daioglou, V., Steinmann, Z.J.N. et al. Biomass residues as twenty-first century bioenergy feedstock—a comparison of eight integrated assessment models. *Climatic Change* 163, 1569–1586 (2020). <https://doi.org/10.1007/s10584-019-02539-x>

Li, W., Ciais, P., Stehfest, E., van Vuuren, D., Popp, A., Arneth, A., Di Fulvio, F., Doelman, J., Humpenöder, F., Harper, A. B., Park, T., Makowski, D., Havlik, P., Obersteiner, M., Wang, J., Krause, A., and Liu, W.: Mapping the yields of lignocellulosic bioenergy crops from observations at the global scale, *Earth Syst. Sci. Data*, 12, 789–804, <https://doi.org/10.5194/essd-12-789-2020>, 2020.

Li, W., Ciais, P., Makowski, D. et al. A global yield dataset for major lignocellulosic bioenergy crops based on field measurements. *Sci Data* 5, 180169 (2018). <https://doi.org/10.1038/sdata.2018.169>

Ramirez Camargo, L., Castro, G., Gruber, K. et al. Pathway to a land-neutral expansion of Brazilian renewable fuel production. *Nat Commun* 13, 3157 (2022). <https://doi.org/10.1038/s41467-022-30850-2>

2. Belowground carbon/soil carbon dynamics are not mentioned at all, although it seems like Harris et al. considered this. I would expect to see a quantification of the soil carbon implications of the land area pledged for CDR, or at least, a solid discussion of what might be expected under different climatic conditions/forest types. In general, I would expect that relative to cropland both afforestation and bioenergy crops may enhance soil carbon stocks, although effects may be heterogenic. I include a couple of references that may help, although I am sure that there are other papers out there.

Bell, S. M., Barriocanal, C., Terrer, C., & Rosell-Melé, A. (2020). Management opportunities for soil carbon sequestration following agricultural land abandonment. *Environmental Science & Policy*, 108, 104-111. <https://doi.org/10.1016/j.envsci.2020.03.018>

Cook-Patton, S.C., Leavitt, S.M., Gibbs, D. et al. Mapping carbon accumulation potential from global natural forest regrowth. *Nature* 585, 545–550 (2020). <https://doi.org/10.1038/s41586-020-2686-x>

Qin, Z., Dunn, J. B., Kwon, H., Mueller, S., & Wander, M. M. (2016). Soil carbon sequestration and land use change associated with biofuel production: empirical evidence. *Gcb Bioenergy*, 8(1), 66-80. <https://doi.org/10.1111/gcbb.12237>

Ledo, A., Smith, P., Zerihun, A., Whitaker, J., Vicente-Vicente, J. L., Qin, Z., ... & Hillier, J. (2020). Changes in soil organic carbon under perennial crops. *Global change biology*, 26(7), 4158-4168. <https://doi.org/10.1111/gcb.15120>

3. Biophysical effects of land-based negative emission technologies will affect the performance of solutions. This must be highlighted, and the argument that it does not fit anywhere in the paper does not hold. Bioenergy crops have been associated with a cooling effect relative to cropland, although effects will vary based on location (see e.g., Wang et al. (2021), Wang et al. (2023), Muri (2018)). Afforestation in the tropics has been associated with a cooling effect, whilst for higher latitudes, reforestation warms the winter climate (Windisch et al., 2021). I think especially the latter is important for Russia's NDCs.

Wang, J., Li, W., Ciais, P. et al. Global cooling induced by biophysical effects of bioenergy crop cultivation. *Nat Commun* 12, 7255 (2021). <https://doi.org/10.1038/s41467-021-27520-0>

Wang, J., Ciais, P., Gasser, T., Chang, J., Tian, H., Zhao, Z., ... & Li, W. (2023). Temperature Changes Induced by Biogeochemical and Biophysical Effects of Bioenergy Crop Cultivation. *Environmental Science & Technology*, 57(6), 2474-2483.

Muri, H. (2018). The role of large—scale BECCS in the pursuit of the 1.5 C target: an Earth system model perspective. *Environmental Research Letters*, 13(4), 044010.

Windisch, M. G., Davin, E. L., & Seneviratne, S. I. (2021). Prioritizing forestation based on biogeochemical and local biogeophysical impacts. *Nature Climate Change*, 11(10), 867–871. <https://doi.org/10.1038/s41558-021-01161-z>

4. Should address the need for infrastructure for long-term CO₂ storage somewhere for the cases of DACCS and BECCS. Rosa et al. highlights developing projects in Europe.'

Rosa, L., Sanchez, D. L., & Mazzotti, M. (2021). Assessment of carbon dioxide removal potential via BECCS in a carbon-neutral Europe. *Energy & Environmental Science*, 14(5), 3086-3097. <https://doi.org/10.1039/D1EE00642H>

5. I suggest to put the NDC area pledges even stronger into the context of land use projections in 1.5C scenarios from integrated assessment. IIASAs AR6 database offers detailed data on specific scenarios that could be used to compare NDCs with future land use change for different combinations of Shared Socio-economic Pathways with Representative Concentration Pathways. Also, comparing with geospatial land use projections could offer valuable insights (e.g., Chen et al. (2020) or Hurtt et al. 2020)).

<https://data.ece.iiasa.ac.at/ar6/>

Chen, M., Vernon, C.R., Graham, N.T. et al. Global land use for 2015–2100 at 0.05° resolution under diverse socioeconomic and climate scenarios. *Sci Data* 7, 320 (2020). <https://doi.org/10.1038/s41597-020-00669-x>

Hurtt, George C., et al. "Harmonization of global land use change and management for the period 850–2100 (LUH2) for CMIP6." *Geoscientific Model Development* 13.11 (2020): 5425-5464. <https://doi.org/10.5194/gmd-13-5425-2020>
Data: <https://luh.umd.edu/>

6. I am wondering if the need for policy instruments to support CDR should be highlighted even stronger as a means to tighten the gap between pledges and actual deployment. See e.g., Wähling et al. for the case of BECCS.

Wähling, L. S., Fridahl, M., Heimann, T., & Merk, C. (2023). The sequence matters: Expert opinions on policy mechanisms for bioenergy with carbon capture and storage. *Energy Research & Social Science*, 103, 103215. <https://doi.org/10.1016/j.erss.2023.103215>

7. I cannot see that the Nabuur paper has been referenced, although claimed so in the rebuttal. I agree with reviewer 1's comment and with the message of Nabuurs et al. (2023) that carbon fluxes from unmanaged forests should ideally be reported, and that this point should be discussed somewhere.

Nabuurs, G.J., Ciais, P., Grassi, G. et al. Reporting carbon fluxes from unmanaged forest. *Commun Earth Environ* 4, 337 (2023). <https://doi.org/10.1038/s43247-023-01005-y>

P.2 Lines 18-21. Could also point out that global warming is a driver of biodiversity loss and that land-based climate change mitigation through afforestation or BECCS may help reduce impacts on biodiversity relative to a future with weaker mitigation efforts (see Jordan et al. (2023) and Hanssen et al. (2022)).

Jordan, Cristina-Maria, et al. "Spatially and taxonomically explicit characterisation factors for greenhouse gas emission impacts on biodiversity." *Resources, Conservation and Recycling* 198 (2023): 107159. <https://doi.org/10.1016/j.resconrec.2023.107159>

Hanssen, S. V., Steinmann, Z. J., Daioglou, V., Čengić, M., Van Vuuren, D. P., & Huijbregts, M. A. (2022). Global implications of crop-based bioenergy with carbon capture and storage for terrestrial vertebrate biodiversity. *GCB Bioenergy*, 14(3), 307-321. <https://doi.org/10.1111/gcbb.12911>

P.9 Lines 9-13. This is spot on. Meeting such land use changes at local levels would require major change in policies and local socio-technical conditions, they must be supportive enough, this seems challenging (see Næss et al. (2024)).

Næss, J. S., Henriksen, I. M., & Skjølsvold, T. M. (2024). Bridging quantitative and qualitative science for BECCS in abandoned croplands. *Earth's Future*, 12, e2023EF003849. <https://doi.org/10.1029/2023EF003849>

P.9 Lines 25-43. I think Russia's CDR area pledge should be put in context with historical cropland abandonment (see Lesiv. et al.). In general, abandonment is widespread around the globe, and either letting this land regrow or converting it to bioenergy production/BECCS offers a good CDR potential (see Gvein et al.).

Lesiv, M., Schepaschenko, D., Moltchanova, E. et al. Spatial distribution of arable and abandoned land across former Soviet Union countries. *Sci Data* 5, 180056 (2018). <https://doi.org/10.1038/sdata.2018.56>

Gvein, M.H., Hu, X., Næss, J.S. et al. Potential of land-based climate change mitigation strategies on abandoned cropland. *Commun Earth Environ* 4, 39 (2023). <https://doi.org/10.1038/s43247-023-00696-7>

P.14 Lines 11-12. Can you be more specific and inform if this is dedicated planting of trees or natural regrowth, or both?

Table S1. Although Harris et al. seem convincing, some of these carbon dioxide removal factors may seem high to me. It is somewhat unclear if natural regrowth is relied on or if it also involved tree planting. Cook-Patton et al. provides an overview of natural regrowth rates in different climate zones, and they seem somewhat lower. I think some more comparisons with other literature could be beneficial to provide an indication of if there is a variation in reported values or not.

Best regards,
Jan Sandstad Næss

Version 2:

Reviewer comments:

Reviewer #4

(Remarks to the Author)

Thank you for the improvements made to the manuscript. The soil carbon sensitivity for selected activities was appreciated. I note that several of my comments were not acted upon, and in some cases solid justification was given. However, other aspects still requires revisions.

The results presented here are very similar to the Land Gap Report. Several figures contain similar data. More care should be taken to provide citations to the Land Gap report where there is clear overlap, including for data in individual figures. Examples include figure 2 and figure 3 in the manuscript. E.g., figure 1 in the 2023 land gap report that shows land required for CDR in national climate pledges, which is a different way to visualize the data shown in Figure 3 in the submitted manuscript. It even states some of the same/similar country-shares (for example, Russia 35%, US 12%, Saudi 20%). Figure 3 in the 2023 land gap report is like Figure 2 in the submitted manuscript (both figures include data on land requirements and number of countries).

https://landgap.org/downloads/2022/Land-Gap-Report_FINAL.pdf
https://landgap.org/downloads/2023/Land-Gap-Report_2023-Briefing_FINAL.pdf

BECCS calculations still have major improvement potential. It is too simple for a paper addressing land requirements of CDR. Using a global average BECCS carbon removal rate based on LPJ-Guess to quantify national-level BECCS land requirements does not follow state-of-the-art methods, considering all the spatial yield data that is available from multiple sources (including LPJ-Guess). On top of that, comes opportunities to utilize land-free bioenergy feedstocks (see for example, Wu et al.). I agree that a change in approach is unlikely to affect the main conclusion of the paper (e.g. over-reliance of land in pledges) considering the importance of reforestation and restoration, but it will affect sub-results and sub-findings that support the main conclusion including all BECCS results. Currently, the national-level results and land requirements for the five countries with BECCS pledges has low value, perhaps especially important for the US pledge. It still needs revision.

Wu, F., Pfenninger, S., & Muller, A. (2024). Land-free bioenergy from circular agroecology—a diverse option space and trade-offs. *Environmental Research Letters*, 19(4), 044044.

Using nation-specific yields would already help. As noted below, it makes more sense to consider second generation energy crops in 2050 than first generation. And a quantitative basis for the discussions on implications of different bioenergy feedstocks on land requirements should be provided.

I note that recently published research has provided scenario analysis of the land use implications of the NDCs for a SSP2 scenario (SSP2-NDC). I think that the insights provided there considering how NDCs affects future land use is important and merits a mention here. E.g., in SSP2-NDC increased forest cover towards 2060 comes at the expense of rangeland and other natural land (nonforested ecosystems, shrublands, deserts). Also note that the net land use change towards forests and BECCS seems lower than the land requirements quantified here.

Humpenöder, Florian, et al. "Food matters: Dietary shifts increase the feasibility of 1.5° C pathways in line with the Paris Agreement." *Science Advances* 10.13 (2024): eadj3832. <https://doi.org/10.1126/sciadv.adj3832>

Some specific comments:

Please note that line numbers refer to the manuscript version without track changes.

P.2 Lines 21-24, and several other places in different sections in the manuscript addresses/discusses impacts on biodiversity. CDR may help hinder irreversible biodiversity losses caused by increasing temperatures that leads species niche limits to exceed (see Trisos et al.). The manuscript does a very good job in highlighting the sustainability risks of land use change. I think there is wide agreement that CDR should not be used as an excuse to avoid emission cuts (e.g., p9 lines 41-47). However, the lack of a discussion of the wider benefits of achieved land-based mitigation from CDR on the same sustainability indicators makes the communication too one sided (see Hirata et al. for a thorough analysis of land-based CDR biodiversity implications). I'd like to see some more nuance in the discussions.

Hirata, A., Ohashi, H., Hasegawa, T. et al. The choice of land-based climate change mitigation measures influences future global biodiversity loss. *Commun Earth Environ* 5, 259 (2024). <https://doi.org/10.1038/s43247-024-01433-4>

Trisos, C.H., Merow, C. & Pigot, A.L. The projected timing of abrupt ecological disruption from climate change. *Nature* 580, 496–501 (2020). <https://doi.org/10.1038/s41586-020-2189-9>

P.5 Lines 18-26. The simple approach of using a global average rate for BECCS is inconsistent with the attempt to capture some locality that is done for restoration and reforestation. It also disregards a lot of work done to spatially model energy crops over the last couple of decades, including using the LPJ family of models (LPJ-Guess mentioned and cited in-text). The approach used here does not follow state-of-the-art. It is something that could be easily improved, and that also should be improved. Chosen approach might not be expected to change your main conclusion (e.g., over-reliance on land in climate pledges), but it does affect sub-results and sub-findings that supports the totality. A key example is United States and results in Figure 3. I re-iterate the need to improve the modelling of bioenergy crops and to capture effects of locality for the five countries with BECCS pledges.

P5. Lines 19-22. As a more concrete example, you could for example point to that Switzerland and UK could meet a share of their BECCS pledges with no land requirements by implementing CCS in incinerators (waste-to-energy, also true for other countries). See Rosa et al. for a quantification, I suggest comparing emission pledges to country-specific potentials.

Rosa, L., Sanchez, D. L., & Mazzotti, M. (2021). Assessment of carbon dioxide removal potential via BECCS in a carbon-neutral Europe. *Energy & Environmental Science*, 14(5), 3086-3097.

P.5 Lines 27-30. It is unclear why bioenergy should be separated into energy sector pledges and BECCS pledges. Cannot these pledges perfectly align, as BECCS is a multi-functional process which both produces energy and delivers negative emissions? In some cases, as for wastes (for example, CCS in incinerators or biomethane production with CCS), it may even serve at least three functions, including waste treatment, energy production, and delivery of negative emissions.

Note that there are 119 ongoing CCS projects under development in Europe, several which involve incinerators.

Levina, E., Gerrits, B., & Blanchard, M. (2023). CCS in Europe – Regional Overview. Global CCS Institute. <https://www.globalccsinstitute.com/resources/publications-reports-research/ccs-in-europe-regional-overview/>

P5 lines 23-24 and P11 lines 14-15. Does Krause et al. (2019) that you rely on for bioenergy capture rates from LPJ-Guess specifically address bioenergy and first-generation energy crops (edible feedstocks)? I see no clear indication of it, neither in the article, in their SI, or in the data published on Figshare. Did I miss something, or is this the wrong citation (is it really Krause et al. (2018)?)? I guess Krause et al. (2019) quantifies NPP of some plant functional types, but these were probably not parameterized as typical bioenergy feedstocks. Please clarify how Krause et al. was used.

Krause, A., V. Haverd, B. Poulter, P. Anthoni, B. Quesada, A. Rammig, and A. Arneeth. "Multimodel Analysis of Future Land Use and Climate Change Impacts on Ecosystem Functioning." *Earth's Future* 7, no. 7 (July 2019): 833–51. <https://doi.org/10.1029/2018EF001123>.

Krause, A. et al. Large uncertainty in carbon uptake potential of land-based climate-3 change mitigation efforts. *Glob Change Biol* 24, 3025–3038 (2018).

As all the BECCS pledges are for 2050 (according to land calculator), it is surprising to rely on first-generation crops and not second generation. Daioglou et al. highlights that lignocellulosic crop (including miscanthus, willow, and eucalyptus) dominates future bioenergy supply in IMAGE scenarios, whilst edible crops play a minor role (see fig. 7). The same also happens in other models. Also, unclear why irrigated crops were chosen and not rain-fed, especially since water scarcity risks were previously highlighted.

Daioglou, Vassilis, et al. "Integrated assessment of biomass supply and demand in climate change mitigation scenarios." *Global Environmental Change* 54 (2019): 88-101. <https://doi.org/10.1016/j.gloenvcha.2018.11.012>

As an example on the importance of crop types, Li et al. predicts with machine learning techniques a global mean yield of 16.3 tDM ha⁻¹ yr⁻¹ for lignocellulosic crops, which may correspond to about 8 tC ha⁻¹ yr⁻¹ (or sequestered 30tCO₂ ha⁻¹ yr⁻¹) harvested. If 90% of this is captured in a thermal power plant, then this is nearly 3x higher removal rate than the value used in your work (10.1 tCO₂ ha⁻¹ yr⁻¹, Table S1) from Krause et al 2019 and way outside the ranges used in the uncertainty analysis.

Li, Wei, et al. "Mapping the yields of lignocellulosic bioenergy crops from observations at the global scale." *Earth System Science Data* 12.2 (2020): 789-804. <https://doi.org/10.5194/essd-12-789-2020>

P5, lines 25-26. And IAMs show lower yield values for second generation bioenergy crops in comparison with the random forest model from Li et al. cited above (see fig 5). IAM yield maps (IMAGE, Magpie, GLOBIOM) also underestimate yields relative to field observations (see fig 6, Li et al.). Are you sure it is right to "give the impression" that IAMs rely on too high energy crop yields?

Also, text says there is more information in SI, but I did not find any.

P.5 line 25. IAM abbreviation not spelled out previously, should remove abbreviation and spell out. Might need to spend a few words describing what these models are as well.

P6, lines 11.13. According to Bluwstein & Cavanagh, the land acquisition peak in 2011 was about 12.4 Mha yr⁻¹, not 7 Mha yr⁻¹. See fig. 10. Also, should specify that this is Global South only, and not worldwide. If a mean is taken over a period around 2011, then 7Mha could be right, but then please specify the year range. Strongly recommend comparing against remotely sensed data in addition, see for example the Hilda+ paper (Winkler et al.). It shows annual changes in the 2000s between 6-11 Mha yr⁻¹ (see fig3).

Bluwstein, J., & Cavanagh, C. (2022). Rescaling the land rush? Global political ecologies of land use and cover change in key scenario archetypes for achieving the 1.5 °C Paris agreement target. *The Journal of Peasant Studies*, 50(1), 262–294. <https://doi.org/10.1080/03066150.2022.2125386>

Winkler, K., Fuchs, R., Rounsevell, M. et al. Global land use changes are four times greater than previously estimated. *Nat Commun* 12, 2501 (2021). <https://doi.org/10.1038/s41467-021-22702-2>

A follow up question to this. How good of a proxy is land acquisition / transactions as a land-use change indicator? My guess is that it is far from perfect. Re-iterate the need to consider remote sensing products.

P7. Line 5. 322 million ha is the mean (not specified)? Also, across what scenarios? SSPs in combination with RCP-1.9?

P7 Line 8. Provide the full range here as well? The complete feasibility space is important. These pathways should not be viewed as a statistical sample (see Huppmann et al., box 1).

Huppmann, D., Rogelj, J., Kriegler, E. et al. A new scenario resource for integrated 1.5 °C research. *Nature Clim Change* 8, 1027–1030 (2018). <https://doi.org/10.1038/s41558-018-0317-4>

P.7 lines 46-48. Refers to the grain-for-green programme? Perhaps, it should be mentioned directly.

P.9 Lines 18-19. This depends on where forest expansion happens. Too unspecific, please spend some more words to explain.

P9. Line21-22. Should differentiate between increases in forest cover for carbon removal and increases in land area for energy crops. This statement is only right for forest cover, not for energy crops (see page 7, lines 3-9). These two land uses do not serve the same function (BECCS can be used both for energy and CDR).

Page 9. Lines 41-43. Land requirements of future food production or biodiversity impacts have not been modelled here quantitatively, so how robust is this conclusion? As it stands now, it seems like qualitative speculation with insufficient support. Note that in contrast Xu et al. highlights that delaying implementation of land-based mitigation measures may threaten food security due to feedback loops on global warming. Some comparisons with quantitative studies may be needed to make this conclusion (planetary boundaries? Hirata et al. cited above?). Should have a look at Humpenöder et al. fig 6 showing some expected loss of natural area in SSP2-NDC.

Xu, S., Wang, R., Gasser, T. et al. Delayed use of bioenergy crops might threaten climate and food security. *Nature* 609, 299–306 (2022). <https://doi.org/10.1038/s41586-022-05055-8>

Humpenöder, Florian, et al. "Food matters: Dietary shifts increase the feasibility of 1.5° C pathways in line with the Paris Agreement." *Science Advances* 10.13 (2024): eadj3832. <https://doi.org/10.1126/sciadv.adj3832>

Also note that many scenarios with major expansion of land-based mitigation involves agricultural intensification and a decrease in pasture area to free land for mitigation purposes. The latter relies on dietary shifts. This strategy can help avoid sustainability impacts.

P10. Lines 3-5. The justification for this statement ("unrealistic targets") is to my impression primarily based on previous land use change rates. This argument needs to be repeated here again for more clarity.

P.11 Line 35. Specify Table S2?

P11. Lines 30-37. How have you accounted for uncertainty in FAO reported forest extent for indirect pledges? Is it included in the error bar shown in Fig 1b? This did not become clear to me after checking the land calculator either. FAO data on forests generally seems somewhat disputed, Lesiv et al. and Bastin et al. both show higher forest cover compared to FAO FRA data.

Lesiv, M., Schepaschenko, D., Buchhorn, M. et al. Global forest management data for 2015 at a 100 m resolution. *Sci Data* 9, 199 (2022). <https://doi.org/10.1038/s41597-022-01332-3>

Bastin, Jean-François, et al. "The extent of forest in dryland biomes." *Science* 356.6338 (2017): 635-638.

Fig 1. A land use change for bioenergy/BECCS is not equal to reforestation. Please improve visualization and differentiate. I did not find the explanation of indirect pledges very clear in the caption, perhaps spending some more words there to describe it could help. I suggest to also put some more information in the graph by stacking the bars and separating into continents or similar, although this might be a more subjective recommendation.

Fig. 4. Same comment as before, that bioenergy/BECCS is not reforestation (at least, affecting UK). Caption is not informative enough to understand it for a reader with limited time skimming through. Suggest providing some more detail.

For BECCS, it is unclear what carbon losses are assumed throughout the supply chain, what carbon capture efficiency is assumed, and how you deal with any supply chain leakages of CO₂. Capture efficiency will vary a lot between conversion pathways such as bioelectricity, biofuel, biomethane, etc. This directly affects BECCS land use.

All the BECCS pledges are for 2050, but it is unclear what background climate was used to produce LPJ-GUESS yields. Expected effects of climate change on yields and consequently the land requirement should be highlighted.

The same question regarding impacts of climate change on forest growth. Harris et al. studied the past period of 2001-2019. How should climate change be expected to affect removal factors and quantified future land area requirements from emission pledges? Especially important for Russia's major emission-based pledge in 2050?

As a final note, I would have appreciated if you could have helped me as a reviewer (with limited time available) out by referring in the rebuttal specifically to the lines where you made changes in the manuscript or by quoting the change in the rebuttal. As a reviewer, my key interest is to see what improvements were made in the manuscript, not only the response to comments.

Jan Sandstad Næss

Version 3:

Reviewer comments:

Reviewer #4

(Remarks to the Author)

The authors have been very responsive to my comments, both by improving the paper and providing clarifications. I include some further reflections from my side that should be straightforward to address. Line numbers refer to the clean manuscript version.

I am overall happy that the comment on bioenergy feedstock, yields, and efficiencies was taken seriously. There is now more transparency considering the relative importance of yields and what is here termed conversion efficiency (combination of carbon capture efficiencies, and other losses). The choice of a 60% conversion rate for the main results shown comes across as reasonable, as this is somewhere in between what may be expected across different conversion pathways such as for bioelectricity, biofuels, biogas, etc. Thank you for the increased clarity.

I found the table that was included in the rebuttal on removal factors to be very useful as it exemplifies the effect of feedstock, chosen models/observations and conversion efficiency, and I recommend to include it also in the supplement to ensure coverage of a larger feasibility space. Perhaps with additional columns in the table indicating feedstock and effects on quantified land use (hectare). It is important in order to understand how results are affected by subjective choices. In fact, should probably even include another lower conversion efficiency in the range of 40-50% representing a liquid biofuel pathway (such as Fischer-Tropsch diesel), see for example Hanssen et al. Table S2 for reviewed CCS efficiencies. Likewise, high-end performing bioelectricity pathways achieving 80-90% conversion efficiency is a part of the feasibility space and should in my opinion be shown as a minimum in an SI. It is definitely not about making excuses for government policies, but rather to inform about the feasibility space and being rigorous.

Hanssen, S.V., Daioglou, V., Steinmann, Z.J.N. et al. The climate change mitigation potential of bioenergy with carbon capture and storage. *Nat. Clim. Chang.* 10, 1023–1029 (2020). <https://doi.org/10.1038/s41558-020-0885-y>

As also stated by the authors, 2nd generation energy crops have recently been implemented and parameterized in multiple DGVMs, fully coupled ESMs and similar frameworks. Thus, if country-level or biome yield data for energy crops from DGVMs or similar can be obtained I would support their use (but not a requirement). See for example:

Stenzel, F., Greve, P., Lucht, W. et al. Irrigation of biomass plantations may globally increase water stress more than climate change. *Nat Commun* 12, 1512 (2021). <https://doi.org/10.1038/s41467-021-21640-3>

Upgraded LPJmL5 version https://www.negemproject.eu/wp-content/uploads/2021/06/NEGEM_D3.1.pdf

Cheng, Yanyan, et al. "A bioenergy-focused versus a reforestation-focused mitigation pathway yields disparate carbon storage and climate responses." *Proceedings of the National Academy of Sciences* 121.7 (2024): e2306775121.

Li, W., Yue, C., Ciais, P., Chang, J., Goll, D., Zhu, D., Peng, S., and Jorner-Puig, A.: ORCHIDEE-MICT-BIOENERGY: an attempt to represent the production of lignocellulosic crops for bioenergy in a global vegetation model, *Geosci. Model Dev.*, 11, 2249–2272, <https://doi.org/10.5194/gmd-11-2249-2018>, 2018.

Ai, Z., Hanasaki, N., Heck, V., Hasegawa, T., and Fujimori, S.: Simulating second-generation herbaceous bioenergy crop yield using the global hydrological model H08 (v.bio1), *Geosci. Model Dev.*, 13, 6077–6092, <https://doi.org/10.5194/gmd-13-6077-2020>, 2020.

Melnikova, I., Ciais, P., Tanaka, K. et al. Relative benefits of allocating land to bioenergy crops and forests vary by region. *Commun Earth Environ* 4, 230 (2023). <https://doi.org/10.1038/s43247-023-00866-7>

Also, it can be observed that several, or most, of the studies do compare predicted yields with observations. Regional performance varies across models. Some of them may perform better in the US than JULES (Littleton et al.).

P6. lines 7-9. I read the response, but it still did not become clear why this disaggregation in energy and BECCS pledges should be used as a justification to assume that other bioenergy demand outside of BECCS should be met with wastes and residues. While I agree that non-BECCS land usage for bioenergy doesn't need to be modelled here, the statement draws an artificial boundary between biomass feedstock for bioenergy and biomass feedstock for BECCS that does not necessarily exist considering BECCS multifunctionality. CCS can be implemented in a variety of conversion pathways relying on different feedstocks. I suggest to delete the second part of the sentence, e.g. "assuming that bioenergy demand outside of BECCS could be met with wastes and residues". Also suggest to instead simply restate that the focus is on CDR and not energy as the reason to not model bioenergy land requirements (or an alternative could be to just delete lines 7-9 altogether). Otherwise, I find the text already added on the potential contribution from wastes in Europe sufficient (and useful!).

P11. lines 20-21. I need to repeat a comment for new consideration, as I don't think the message came through and the wrong sentences were quoted. I'll try again, being more clear. Here in page 11 it is stated:
"It is alarming that the extent of land required for CDR in government climate pledges already tracks against the upper end of mid-century scenario expectations."

I note from Table 2 that land use change for reforestation is quantified as 450 Mha and for BECCS as 61 Mha.

Then in page 8, the following is stated:

"Modelled pathways that limit warming to 1.5°C with no or limited overshoot show increases in forest cover for carbon removal of 322 million ha (median, with a range of -67 to 890 million ha)¹⁶. Many of these pathways also include large amounts of energy cropland area, to supply biomass for bioenergy and BECCS, with 199 (median, 56-482 range) million ha by 2050."

450 Mha of land use change for reforestation is indeed above median and in the upper range/half in 1.5C pathways (although, still half of 890 Mha in the far high-end). However, 61 Mha for BECCS is far below median and bordering the lower end of the 56-482 Mha range (although range also include land use for bioenergy in addition to BECCS, and shares between the two vary across models (see Daioglou et al. (2020) for EMF-33 energy results, fig1)).

Considering that BECCS land requirements is only about one quarter of the median in 1.5C pathways and borders the low end of the range, I therefore propose to differentiate between land use change for increased forest cover and energy crops in p11 lines 20-21. Otherwise, the statement seems misleading.

Daioglou, V., Rose, S.K., Bauer, N. et al. Bioenergy technologies in long-run climate change mitigation: results from the EMF-33 study. *Climatic Change* 163, 1603–1620 (2020). <https://doi.org/10.1007/s10584-020-02799-y>

fig 1. BECCS was separated, but now it is unclear what "conditional" and "unconditional" pledges in the legend means. While these two terms may be established elsewhere, they are not used anywhere else in this manuscript (I tried to search). Adding an explanation here would be helpful. Also, with current figure design, seems like BECCS fits neither conditional or unconditional? Is that correct, or should BECCS also be separated (BECCS-conditional and BECCS-unconditional, or only one of the two)?

fig 3. I noticed that the source data given in column B in ("455217_3_data_set_9406687_shhr29.xlsx", sheet "Figure 3 share of global land") differs from the labels given in the figure next to country names. Fix?

P.13 lines 11-13 refers to the SI for details on bioenergy country yield calculations. In the SI, in addition to providing some country-numbers (Table S1) the following is written: "The yield uptake rates and SD for bioenergy were taken from Li et al 2020, with conversion efficiencies applied to yield uptake following Vaughan et al 2018".

This short explanation is not very transparent and makes the results difficult to replicate. How did you go from gridded data from Li et al. to country-specific yields? Not sure what was done, but a decent proxy could be to use gridded data of current cropland cover to filter and weight yield data. Also, what does the standard deviation represent, is it calculated based on spatial yield variability within the country? Do you include conversion efficiency in the SD as well? My impression is no, and if so, I really think you need to include in the SI the table from the rebuttal describing effects of yields and conversion efficiencies to better cover the feasibility space.

p.11 41-47. Some of these claims are still very strong with limited quantitative support from the analysis that was done. It has indeed been shown here that some large countries that rely on petroleum production have made major CDR pledges with associated land use. I am however less convinced that it has really been shown that pledges push mitigation burden onto the land sector instead of fossil fuel phase out. Although it probably is right, there is no quantitative basis here provided on emission cuts supporting the statement. Same regarding biodiversity and food security beyond land use indicators. I note that some text was shuffled around but that the message seems the exact same as in the previous manuscript version. The point made that taking land out of production in Global South may compromise food security for local populations is valid considering the geographical distribution of pledges shown, but perhaps it should be written in directly to support the claim. In contrast, I feel like the claims made in the abstract is better balanced.

Response to review – Over-reliance on land for carbon dioxide removal in national climate pledges

*NB: line numbers referred to below are in the track change version, with deletions in text.

Reviewer #1:

Key results The authors reviewed country pledges that express their climate commitments related to land and that are represented as a range of different metrics and qualitative ambitions. Several countries (53 of ~165, or ~32%) did not provide enough information for the authors to include in their assessment. This compilation exercise by the authors is in itself commendable, as the information provided by countries tends to lack transparency, making it difficult to synthesize the information as was done in this paper. The authors’ recommendation in the conclusion is well supported that greater transparency is needed around the approach to land management in climate mitigation plans and the assumptions made more clear about the land area needed for their land-based mitigation commitments. From this review, the authors conclude that based on current pledges, approximately 1.1 billion ha of land would be needed globally for land-based CO2 removals to be delivered as pledged by countries over the time period 2020-2060, an area equivalent to two-thirds of global cropland area. From this analysis, the authors suggest that these commitments place too much expectation on land to deliver on the Paris goal of achieving a net balance between anthropogenic emissions by sources and removals by sinks of GHGs in the second half of the century, potentially undermining the need for near-term emission reductions.	We thank the reviewer for these remarks
Validity and robustness of conclusions	
The type of analysis as conducted in this paper is helpful, but mainly as an example of how the type of analysis the authors want to conduct in a credible manner is simply not possible based on the information provided. I would like to see it re-written with this angle, rather than trying to pass off the analysis that was done as something analytically robust. As currently written, it is more appropriate for a specialized policy audience as a back-of-the-envelope estimate for those following UNFCCC negotiations. As currently formulated, the estimates lack the analytical rigor I would	#1 We thank the reviewer for this comment. We have reframed the abstract, introduction and objectives at the start of the paper in many places to make clearer that the land found in climate pledges may or may not be implemented. We make two specific points related to this:  - That we assess how much land would be required if country pledges were implemented, not how much land we expect to see used for CDR in reality. See Page 2, line 45 – page 3, line 3.

expect from a high-profile Nature publication and the findings do not represent a particularly significant or novel advance to specialists in the field. This is less the fault of the authors and more the fault of the bad data they had to work with.

- That the information given by countries in their climate pledges is of insufficient detail to provide accurate estimate of the amount of land that would be required to meet climate mitigation pledges. See Page 3, lines 19-26.

We also underscore this point in the discussions and conclusion where we talk about more transparency needed.

However, we defend the analysis as being analytically robust on the basis of the detail we have gone into (where possible) to quantify land area in pledges. This should be clearer in the spreadsheet now that our calculation assumptions have been added (see column T). For some countries we have separately assessed up to 6 different land sector activities in order to quantify the different carbon uptake potential of different activities (eg: Niger, Uganda, Uruguay). Other countries have clearly stated how much land they intend to use for mitigation (eg: Brazil, Bolivia, Colombia). For countries that have only stated tonnes CO² removed, there is uncertainty associated with the assumptions of activity types and different removal factors – we have further emphasised these uncertainties throughout the paper and in the online methods.

In addition, we have added a sensitivity analysis where we assume global average removal factors for all emissions-based pledges (removing any interpretation about the type and location of land-based activities) which shows an increase of 10% from our results, demonstrating it is not our assumptions regarding activity types or removal factors which significantly drive the results.

As the results are an aggregate of a relatively large data-set (194 countries and 296 separate spreadsheet quantifications), differences in interpretations of country intentions where these are unclear do not significantly change the overall results.

The main push of the paper is to voice concern over the amount of land required to remove carbon dioxide at the scale and pace countries have pledged as a way to get us out of this global climate mess. To put this in context, it could be helpful if the authors mention country pledges on the emission reductions side as well, not just for land but overall across sectors; are these equally concerning or is the concern limited just to CDR on land? Are emission reduction pledges more realistic than the land CDR pledges or similarly unrealistic? Might be worth making a point up

#2

We agree with the reviewer that readers should understand the overall context of pledges, and whether pledges on the emissions reductions side are on track and have made several edits to accommodate this.

We quantify land area to highlight that embedded in the already insufficient national climate pledges are unrealistic claims to land that render them even more insufficient. This is an important point in its own right - regardless of how one interprets the possible roles of pledges. We have clarified the role of these pledges as

front that these commitments/pledges may be set by countries to be overly ambitious on purpose, to be used more as aspirational targets/tools for political posturing than as having any basis in reality. The example of corporate commitments made in 2014ish to end deforestation by 2020 comes to mind: was that goal achieved? No. Was it a useful pledge to rally around a collective vision of hope? Yes. Once that narrative is set up about what role these pledges play, and how the Paris Agreement encourages ratcheting up of ambition over time, then the authors could make the recommendation/conclusion backed by their analysis that pledges should be more realistic and transparent about the additional climate mitigation that CDR from the land sector could actually provide.

government commitments under the Paris Agreement (Page 2, lines 27-43), so they are not comparable to voluntary corporate pledges. We have presented a great deal of new data in this paper by focusing on land area as a quantification metric, and believe that comparing these land pledges to emissions reductions in other sectors on a country by country basis is outside the scope of this paper.

To contextualise the land pledges in comparison to pledges in other sectors, we have:

- Made clear the political role that pledges play as aspirational targets, rather than as precise descriptions of what will happen in the future (Page 2, line 45).
- Added a reference to recent assessments from the UNFCCC and UNEP which conclude that both NDCs and LT-LEDS pledges are inadequate, showing that emission reduction pledges are not on track to meet the goals of the Paris Agreement (See Page 2, lines 31-32).
- Added a discussion on estimates that considering 2050 pledges could put us on a path to below 2C, yet given the majority of large land area pledges are in 2050 targets, this calls into question whether these long-term pledges can be implemented (Page 11, line 15-18).
- Made clear that the level of detail in pledges does not allow for accurate assessments of land area (Page 3, lines 19-24).

We agree with the reviewer that it is important to set up a narrative about the role that pledges play and the need to ratchet ambition over time. This is spelt out in our conclusion section where we call for “scaling up ambition levels in near-term emissions reductions rather than striving to achieve what appear to be unrealistic targets for land-based CDR.”

The authors acknowledge that their analysis required many assumptions to arrive at their estimates. Based on my review of the online methods and the excel spreadsheet, these assumptions are not as clearly outlined as I hoped they would be for a reviewer with limited time. More straightforward explanation would be helpful about how the land area estimates per country were calculated and more explanation provided in what assumptions were used where. The authors should clearly acknowledge the limitations of their analysis and that the result from their uncertainty analysis is surely an underestimate.

#3
We have implemented several changes to clarify the assumptions we made in quantifying land area from pledges and to acknowledge the limitations of the analysis. These are:

- Clarified the three methodological approaches we used to calculate land area (direct area, indirect area, emissions based) where this is first mentioned in the manuscript (Page 4, lines 2-8). These approaches are also referenced in the label to Fig 1, and explained in full in the online methods and repeated in the SI, but we agree it is also

	important to make the quantification approaches clear where readers first encounter this.  - We have rephrased the discussion section where we explain why our analysis is likely to be an underestimate as a limitations section, to make clearer the limitations of our study, including not assessing bioenergy (see response #9, below) (Page 10, line 23 – page 11, line 7). - We note that the calculated uncertainty values are an underestimate as they are only based on removal factors, when there are many other areas of uncertainty in the data (in online methods, page 14, lines 20-23). - We have added notes for each calculation row into the spreadsheet to explain the assumptions behind interpreting each land activity from country pledges (see column T). We have added more information to the Notes page of the spreadsheet to explain which spreadsheet columns key information is found (land area results, calculation notes) and also to explain again the direct, indirect or emissions based methodological approach and where in the spreadsheet this information can be found.
Data and methodology	
The authors' division of land into reforestation involving a change in land use vs. restoration of degraded forests is somewhat puzzling to me. The authors imply that reforestation involves a land-use change, but depending on country context reforestation may be synonymous with restoration of forest land remaining forest land. For example, are trees growing back after harvest – as may be prevalent in three of the four countries identified that contribute substantially to the global total land area for CDR (Russia, US and Canada) - counted by these countries as reforestation or restoration of a degraded forest? Regrowth of young secondary forests and plantations may not result in a change of land use because it may be considered by countries as forest land remaining forest land.	#4 Our assumptions regarding reforestation = land use change and restoration of degraded forests does not equal land use change are based on IPCC LULUCF accounting guidelines. Hence, if an NDC or long-term pledge indicated trees growing back after a harvest (such as improved forest management), we would interpret this as forest land remaining forest land, in line with IPCC guidelines, and therefore would categorise this as restoration, as the reviewer rightly points out. In most cases, there is very little information in the pledges, but we used key terms to categorise pledges into our 7 activity types, which then correspond to restoration or land use change (see Table 2, page 6 for CDR typology and terms). To make our search and classification approach clearer, these key terms have now been included in the online methods (page 13), and we have highlighted country examples in Table 1 that use these key terms. We only categorise activities as reforestation if afforestation, reforesting or establishing plantations is specifically mentioned, with no reference to forest management or an existing forest. We have added a clarification that any reference to forest management is considered restoration and regrowth of young secondary forests and plantations may be considered by a country as forest land remaining forest land, but if the pledge does not

	specify the reforestation is taking place in an area categorised as forest land, then we will assume it is land-use change (See Page 5 line 11- page 6, line 3). We accept that our categorisations are somewhat arbitrary, but they are designed to give an indicator of the type of land-use activities pledged by countries and not an accurate prediction of what will happen, as noted in comments above. In the country specific cases the reviewer mentions, our interpretation of activity and land-use change or restoration should now be clear from the notes we have added to the spreadsheet (column T). For the US, Canada and Russia:  - the US refers to opportunities for reforestation and targets for CDR such as BECCS or DACCs, hence we assume all of this would require land use change (we calculate reforestation and BECCS potential separately and do not count the sink capacity of existing forests that they reference). - Russia refers to ‘managed ecosystems’ hence we interpret that as restoration, not land-use change, and they quantify the sequestration potential, which we convert to land area. - Canada refers to increasing its LULUCF sink, so we assume that is in managed forests hence restoration. We exclude the current sink in existing forests. They also refer to BECCS, which we calculate as land-use change. These assumptions and calculations can now be seen for each country in column T of the spreadsheet.
It was difficult for me to follow the specifics of what was happening in the cells of the Land Gap Calculator spreadsheet. It would be helpful to include more information in the Notes tab, and/or provide further explanation in the supplementary Word doc, of a worked example of how estimates are derived for different scenarios when carbon removal commitments were not expressed directly as land areas. For example for each of the pledge types, provide an illustrative example of how the authors translated it into an estimate of land area. Saudi Arabia was based on a pledge on number of trees – the conclusions of the paper depend heavily on how that number of trees estimate was translated into an area of land and what assumptions were used. Currently all that is hiding in spreadsheets and supplementary material, but it’s critical to elevate the	#5 We appreciate this comment from the reviewer, and the difficulty of following the specifics in the Land Gap Calculator spreadsheet. We have now added in further information in the notes tab of the spreadsheet, as requested. This gives specific and relatively detailed guidance for how each pledge was interpreted and calculations made. Table 1 in the manuscript is designed to provide information on how the different types of pledges are interpreted as different activities, as explained above at response #4. We note that in the case of Saudi Arabia, this is labelled as a ‘direct’ land area pledge in column P of the spreadsheet, which means that they gave a land area for their pledge, not only number of trees (the notes

assumptions used to the main text if the paper's conclusions are to be supported.	tab of the spreadsheet now directs people where to find this information). We agree with the reviewer that a pledge made as number of trees is very difficult to translate to a land area. For this reason, we prioritised direct or emissions-based approaches, only basing our land area calculations on number of trees if no other quantifiable information was available (Indirect pledges also include proportion of country or forest area, which is relatively reliable information). In the end 24 country pledges were quantified via indirect approaches (column P of the spreadsheet) and only 13 of these were based on number of trees as indicated in column H of the spreadsheet where the relevant tree density is given, with source data in the Removals Factor tab. Of these, only 2 pledges based on tree-density are over 1 million ha – South Sudan and Uganda – which reflects 0.3% of our results, hence we have minimised overestimation of land from the tree-density approach. We have also included a sensitivity analysis which calculates all pledges that are not directly stated based on a global average removal factor, which results in a 10% increase in land area.
Similarly, Russia's importance to the overall geographic distribution of land in climate pledges (33% of the total) would be particularly important to understand how the estimate was derived, since it was not a direct pledge of land area. How does a pledge to "more than double the absorptive capacity of managed ecosystems" translate into the assumptions made by the authors to arrive at the land area needed? These country pledges may be designed to be vague on purpose, and that point could be made in the paper.	#6 We have added our calculation notes to the spreadsheet, which answers the reviewer's question. Russia's pledge states: "the absorptive capacity of managed ecosystems is expected to increase from the current 535 million tons of carbon dioxide equivalent to 1,200 million tons of carbon dioxide equivalent in forestry", which is now included in column T of the spreadsheet. On page 8 we characterise this as "more than double", but Russia's pledge is emissions-based, meaning the steps to quantify the required land area were relatively straightforward based on default removal factors for Old Secondary forest in the boreal biome, and can be seen in the spreadsheet. The manuscript text on pages 8-9 is intended as a discussion of the country pledges, not a description of how they were calculated. We also emphasise in the discussion section that country pledges are vague, which underpins our key recommendation that more transparency is needed (page 11, line 9).
Analytical approach	
I suggest the authors review national GHG inventories to understand what level of CO2 removals are reported by countries, to put the CDR pledges into context.	#7 We believe the value add of this paper is in quantifying land area and discussing the implications of CDR pledges in terms of the scale of land that is implied.

	Many countries only report net LULUCF emissions in their GHG inventories, and so data on removals vs emissions is not always available - collating this information for all countries is beyond the scope of this paper but could be valuable for future work, particularly in light of discussions about modifying inventory reporting (see response #9, below). We have focused on quantifying the area of additional land that would be needed for removals to meet future pledges – where possible we have removed the current LULUCF sink as the baseline (See for example US, Canada, EU). For example, Canada states it plans to increase its land sink to 100 MtCO₂ by 2050, while the GHG inventory reports a current LULUCF sink of 7 MtCO₂, and so we quantify the land area for 93 MtCO₂ of additional removals. This information is available in column T of the spreadsheet.
It is not clear to me how the mean and standard deviation of areas of restoration and reforestation were calculated in Figure 1.	#8 The calculation of the mean and SD is explained in the online methods, with further information in the SI. Further clarity on this is available in the ‘source data’ spreadsheet, containing the data and some explanation for all figures, included with this resubmission. We have made a reference to the online methods in the Figure 1 label, so it is clearer where to find this information.
The authors didn’t assess bioenergy demand or quantify pledges for protection of existing forests that would result in emission reductions, or include CO₂ removals from primary forests as these are non-anthropogenic removals included in the terrestrial land sink. Focus is on additional C sequestration specifically. More clarification should be provided on how countries account for current CO₂ removals in secondary forest and plantations, vs. additional CO₂ removals from “restoration” or “reforestation” of secondary forest. See also Nabuurs et al. 2023 (Communications Earth and Environment) who argue that carbon dioxide fluxes from all forest land (managed and unmanaged) need to be recorded by countries in order to help track progress towards global climate targets.	#9 Mitigation involving bioenergy is usually reported in energy sector activities of NDCs, and with the scant detail provided regarding feedstocks and conversion technologies, it would be very difficult to provide any reliable estimate of associated land area. At the risk of overinflating the level of lands countries might rely on, we chose not to include bioenergy. We are also clear that our analysis is focused on land for CDR, and not the avoided emissions from protecting standing forests or substitution effects from bioenergy. We have reframed a section in the discussion to be clear on limitations, including that our analysis does not cover these activities (Page 10 lines 23 – page 11, line 7). In terms of the distinction between the terrestrial land sink and anthropogenic removals - current accounting practices for national inventories are to only report the carbon flux on managed lands, which was introduced as a proxy to capture anthropogenic effects as noted by Nabuurs et al. Their recommendation to move to comprehensive land sector accounting has been discussed for a long time, due to the accounting complexities and loopholes introduced by the managed

	land proxy, and now also due to the discrepancy in global estimates of the land sink between national inventories and climate models, as previously discussed by Grassi et al., 2021 (NCC) and 2023 (Earth Sys. Sci. Data). However, our analysis is not about reporting of terrestrial fluxes in national GHG inventories, but rather about activities pledged for future mitigation in NDC and LT-LEDs. There is a distinction between setting a target (as in NDCs) reporting national GHG fluxes (as in national GHG inventories), and accounting towards a target (as in the ETF under the Paris Agreement). The context of the pledges we are assessing is in terms of what should be counted towards a target, not what should be reported, and Nabuurs et al do not tackle the issue of target-setting in their paper. While the recommendation to move towards comprehensive land sector accounting in inventories is a good one, it has larger political implications in terms of what part of that terrestrial flux should count against countries climate targets? Delving into this complex discussion of terrestrial carbon accounting we believe would overcomplicate the paper and would be very difficult to include in space limitations (Giddens et al 2023 discuss this to some extent). To accommodate the reviewers concerns, we have added a reference to Nabuurs et al., 2023 in the discussion in reference to the need to separate land and energy sector targets in NDCs for greater transparency, as this is what would be required in order to implement the comprehensive land sector accounting this author group calls for (see page 11, line 21).
Uncertainty analysis: Note that removal factors and their uncertainties from IPCC 2019 refinement for temperate forests were revised in a correction published by IPCC in July 2023.	#10 We thank the reviewer for pointing this out. While we are aware of this update, we have not included the revised removal factors, because our source for removal factors is Harris et al 2019 rather than IPCC. The Harris et al data is based on the IPCC, but presents removal factors in tonnes CO2 rather than tonnes dry matter, making them more directly applicable in a policy context. We believe that updating the removal factors for temperate forests would make very little different to the results, and this change will make little difference to results, even though the uncertainty on this particular RF is large (because it is applied to very few calculation rows in our spreadsheet).
This may be beyond the scope of the paper, but some reference could be made to the biophysical processes that may enhance or diminish the	#11

climate effects of carbon released or absorbed from forest biomass (e.g. albedo), particularly if a substantial portion of land-based CDR is expected to come from temperate and boreal regions (Russia, Canada, USA).	We tried to fit this in somewhere, but we feel that this is outside the scope of this paper, and could not find anywhere in the discussion that it naturally fit.
Suggested improvements	
Use active voice (We provide a first estimate” vs. “this paper provides a first estimate”)	#12 Thank you for this recommendation, we have revised to include active voice throughout the paper.
Per Nature guidelines (I think), avoid claims of novelty (“it provides the first assessment of its kind”)	#13 We have made this change as well as others to be consistent with the Nature Communications Guidelines, such as not mentioning our study until the last paragraph of the introduction and removing footnotes.
Abstract and throughout: avoid vague statements like “implying impacts on people and food security”. What kind of impacts?	#14 We have added that the likely impacts are due to people being dispossessed of access to land and land-based resources, see page 1, lines 22-23 This is also described in more detail page 11, lines 29-38 with ample references.
Include in the introduction more context for the general reader of what the Paris Agreement goal is – achieve a net balance between anthropogenic emissions by sources and removals by sinks in the second half of the century. Important to include anthropogenic and to stress the role of forests on the sinks side.	#15 We have added this context into the introduction, see page 2, lines 13-16.
Perhaps worth pointing out that the idea that all countries achieving net-zero within their own boundaries doesn’t necessarily make sense because all countries have a different starting point when it comes to carbon dioxide removals on land. Some developing countries are already net zero/net sinks! So it’s an important point to make that to increase CO2 removals beyond those that exist in the country now, we need to assess the level of ambition in climate pledges and what CDR would be on top of current CDR in the land sector.	#16 Assessing the level of CDR that would be on top of current CDR in the land sector is exactly what we are doing in this study. For the most part, countries make clear in their pledges what action is pledged as new or additional compared to current LULUCF sink or source.
Another point to highlight might be how land use history is likely to play a role in how CDR actually plays out – how much CDR an ecosystem can support may differ from the CDR that countries are pledging to deliver. The way that these policies play out and how land is actually used will depend upon local land tenure as well as other social and economic factors.	#17 We think that the reviewer makes an excellent point here and have included this sentence on page 11, lines 38-39: “The way that these policies play out and how land is actually used will depend upon local land tenure as well as other social and economic factors.”
Clarity/context	

“We present a breakdown of what these removals would look like” – this is too vague, suggest deleting from abstract. More clear: “We present a breakdown of how demands for land would be distributed geographically and over time”	#18 We have edited the abstract to reflect this suggestion.
“For more than half of this area, the pledges envisage the conversion of existing land-uses to forests, while the remaining area is for restoration of degraded ecosystems.” What is a “degraded ecosystem” as defined by various countries? How much overlap is there between existing land uses (several of which are degraded from their natural state) and degraded ecosystems? Not clear how this breakdown was determined after reading through the methods.	#19 Our classification of land use activities into those involving land-use change or those involving restoration (such as managed forest remaining forest) is illustrated in Table 2 is now better explained on page 5, and in the online methods, in response to comment #4. We are using the land use change / restoration divide as a proxy to indicate where CDR may be more or less problematic. It is obviously not exact to each country circumstance and land use history given the lack of information in NDCs and we have underscored the vagueness of pledges and the need for more detail and transparency in mitigation commitments at several points in the paper.
Not clear how land for reforestation/plantations for BECCS is calculated	#20 Countries that included BECCS in their pledges did not state what the biomass feedstock would be, as we note on page 6, line 10. Hence, we assumed forest plantations for the bioenergy feedstock, meaning the calculation to convert tonnes removed via BECCS was the same as converting tonnes removed via plantations (using the relevant biome removal factor for plantations based on the country). We have added a sentence at page 6, line 12 to make this link between assuming BECCS feedstock as plantations and calculating land area for plantations clearer.
Introduction: “many climate mitigation approaches that rely on land, such as large-scale afforestation, threaten to exacerbate rather than address the biodiversity crisis.” Change to “some” approaches, as many climate mitigation approaches that rely on land have positive biodiversity benefits if implemented well.	#21 We thank the review for this suggestion and have implemented this change on page 2, line 19.

Reviewer #2:

The paper, entitled "Over-reliance on Land for Carbon Dioxide Removal in National Climate Pledges", makes a significant and relevant	We thank the reviewer for these comments
---	---

contribution to the climate policy literature, focusing on the assessment of land requirements for land-based mitigation options and the associated risks. As a publication based on the analysis of the "Land Gap Report", it provides a timely contribution to the scientific literature, esp. in the context of next NDCs due in 2025. However, there are several issues that require attention and major revisions before publication.	
1.Relevance of the documents for domestic climate policy making:	
a. There is a flurry of analyses of NDCs and LT-LEDS. What is usually missing is a reflection on the political role of these documents. The lack of submitted LT-LEDS, for example, indicates the low political priority that UNFCCC signatories attach to these reporting mechanisms. It should be emphasised that not only is the implementation gap between actual policy and NDC/LT-LEDS huge, but also that these strategy documents have limited relevance for national climate policies and that these political contexts for specific pledges cannot be identified from these documents. This is not to say that a quantitative analysis of the documents should not be done, but the framing of such an analysis should take into account this limitation arising from the data source of strategy documents that may not fully reflect a country's policy priorities.	#22 We thank the reviewer for this insight. We have reconsidered the framing of the paper and agree that it is treating the climate pledges as factual statements. We have added a new paragraph to the introduction that discusses the political context of the pledges, in terms of presenting ambition that may not be realised (See page 2, from line 45 on). One aspect of understanding how realistic these pledges are, however, is to understand the land area that is embedded in CDR claims, which is the objective of this paper. In addition, we have used more conditional language throughout the paper when talking about land in climate pledges that may or may not be realised.
b. The paper should provide additional interpretation and policy contextualisation, particularly regarding the high numbers from individual countries, such as Russia. This will help to interpret the high headline number reported at the beginning of the paper.	#23 The discussion on page 8 is intended to provide policy contextualisation for the countries with very large land area pledges, but this is very brief due to space limits making it difficult to discuss multiple national contexts. We have expanded this discussion (see page 8-9), but help interpret the headline number we have also:  - Added a sentence to the introduction that draws attention to the small number of countries responsible for the majority of results (Page 3, lines 17-18) - Added the notes regarding calculation assumptions to the spreadsheet (see column T) which shows how the calculations were made for all countries.
2.Methodology:	
a. The methodology used to extract data from the documents should be clarified. At present, the supplementary material lists the sources of the documents and calculations but not the exact wording on which the calculations are based. This	#24 We have clarified how the calculations were done by adding a notes Coloumn (T) to the land calculator spreadsheet which explains the

makes it difficult to trace the commitments and check for consistency and accuracy.	assumptions behind the calculations for each spreadsheet row.
b. The paper currently categorises all types of land restoration as mitigation/CDR commitments. Given that land has been managed for a long time and that objectives have changed (e.g. combating desertification, biodiversity,...), it might be worth reflecting that CDR may not be the primary objective of all the pledges collected here, but could also be a side-effect/co-benefit of an initiative that policy makers had in mind for other reasons - it may not be all about CDR. It would be useful to distinguish between pledges that are explicitly focused on carbon removal and those where CDR is a secondary or co-benefit. This clarification will provide a better understanding of the nature and intent of the pledges.	#25 We have revised the paper to make it clearer that we don't categorise all land restoration as CDR, we have only quantified land area that is included in the mitigation component of country pledges. We do not include in results land that is only included in adaptation pledges, or non-climate restoration pledges. Many countries have made restoration pledges under the Bonn Challenge or other initiatives, or significant land sector policies in the adaptation component of their NDCs, which we don't include. The difference, and likely partial overlap, between climate pledges and other restoration commitments is explained on page 7, lines 4-15. We have made revisions here, and in the introduction and where results are first introduced to clarify that the pledges we assessed are only from the mitigation component of NDCs. The reviewer is correct that some of the mitigation pledges reflect other land sector priorities than CDR (such as agricultural regeneration and food security), but as they have been included in the mitigation component of pledges, the country is indicating they will count the CDR from these activities towards their climate mitigation targets.
c. Related to Issue 1: The use of government documents (instead of NDC/LT-LEDS in the analysis should be explained (why are government documents used in some cases?)) There are significant gaps between NDCs/LT-LEDS and actual national policy making (see also Issue 1). This should be better explained - or focus on only one type of document. Alternatively, the paper could consider focusing on one type of document to provide a more consistent analysis.	#26 We have used NDC/LT-LEDS for all countries but 2 – Saudi Arabia and Kazakhstan. Both of the documents we use for these countries are presidential speeches that announce reforestation initiatives as part of climate plans. They are confirmed and public plans, but not included in NDC/LT-LEDS yet as these are not frequently updated. The information on which pledges are official NDC/LT-LEDS or unofficial pledges in other government documents is in column S. Several entries that had not been updated from unofficial to official are now fixed.
3. Permanence/reversibility of CDRs:	

a) The issue of reversibility and different permanence periods for different types of sequestration is critical, but is not addressed in the pledges and is not explicitly raised in the paper. Given the aim of the paper to inform the review of NDCs, it is important to highlight the challenges associated with relying on LULUCF-based CDR to meet climate goals. The potential measures to incorporate them into CDR policy, such as buffer pools and equivalence discounting, should be raised to highlight that land requirements could be even higher if LULUCF-based CDR is responsibly governed to meet climate goals.	#27 We thank the reviewer for this suggestion and agree that it is an important issue, and is one of the key messages of the Land Gap report work, underlying this paper. We have now added a sentence on this in the discussion with a recent reference (page 10, line 21-22), but due to space constraints cannot expand on this issue in the paper.
In conclusion, the paper makes an important contribution to climate policy by addressing the role of land-based mitigation options in national climate commitments. In order to improve the relevance of the paper, it is recommended that the above issues are addressed in a major revision. By doing so, the paper can provide a more comprehensive and insightful analysis of the issue and have a greater impact on the climate policy discourse	We hope that the revisions outlined above have adequately addressed the reviewers concerns and have improved the paper.

Reviewer #3 :

Reviewer #3 (Remarks to the Author):	
I noted reading this paper and consider it a very worthwhile contribution. The paper evaluates the scale and potential land-use conflicts (and other challenges) for nations to deliver on pledges around LULUFC and BECCS. I addresses and highlights the uncertainty and risk of presenting these land pledges in the absence of robust local assessment and especially spatial analysis. The same risk applies to national pledges as they pertain to energy efficiency, renewable energy expectations, and CCUS (especially storage). This lack of spatial grannularity obscures a multitude of risks and uncertainties when it comes to resource capacities, environmental considerations and local values. In the case of the land sector, the lack of coordination between different environmental conventions is especially problematic. The paper is timely, as we approach COP28, proposals to update NDCS, and the IAMs modelling updates ahead of the 7th Assessment Report.	#28 We thank the reviewer for these comments, we particularly agree that the lack of coordination between reporting on progress towards different environmental conventions is problematic.
Acknowledging that I am not an expert in land sector analysis, the analytical work done appears	#29

robust and uses sound sources. The authors also acknowledge clearly the uncertainties and limitations on the analysis but I agree they have landed at a conservative place.	We have improved the analytical work done in the paper through including the calculation assumptions in the spreadsheet (column T), and through conducting a sensitivity analysis which shows that our choice of removal factors and forest biomes does not significantly drive the results.
The paper is also very well written. I had a couple of very minor comments: On page 6, where the authors write: "For example, the global land rush of the 2000s, which was seen as a great threat to small scale farmers, saw no more than seven million ha being transacted per year." Authors could perhaps add some descriptive context noting the broad readership of Nature Comms. E.g.: "For example, the global land rush of the 2000s for the purpose of industrial-scale agriculture and resource extraction..., which was seen as a great threat to small scale farmers, saw no more..."	#30 Thank you, we have implemented this idea and made other similar small edits.
Also suggest the insertion of "approximatey" or "around" in a number of places, given the acknowledged uncertainty in parameter. e.g. "Saudi Arabia has pledged to plant an additional 40 billion trees in neighbouring countries, equivalent to (around) 200 million ha". Otherwise I recommend for publication.	#31 We have added more conditional language to the abstract and intro to indicate that the estimated land area would be required if these pledges are implemented. We have added language to indicate the pledges from Saudi and others are approximate.

Response to review – Over-reliance on land for carbon dioxide removal in national climate pledges

Reviewer #2:

Dear authors, thanks for the careful revision of paper and considering the points raised by the reviewers. The improved transparency in the methods and the new framing about lack of data/details in the pledges improve the paper.	
Two minor things I'd like to raise: - the direct quote from the PA is not the exact quote (at least not from Art. 4	Thanks for pointing this out, we have corrected the quote.
- I think in the supplementary material are a few notes (eg. U6, Z29) and colour (AF17) that should be deleted before publication	Thanks again, we have cleaned up the supplementary data sheet.

Reviewer #4:

The paper evaluates the reliance on land for CDR in national climate pledges. I generally find this topic to be highly important and the results noteworthy. The article is likely to have an impact on the field. I found results largely supported by data, although I think some revisions are needed. I provide some comments below. Thank you for the interesting read.	
1. BECCS pledges seem unlikely to be forest feedstocks. In general, biomass residues are very important in ramping-up biomass supply at low levels of demand, whilst dedicated bioenergy crops dominate when demand exceeds 100 EJ year⁻¹ (Hanssen et al., 2020). The energy yields of short-rotation bioenergy crops exceed managed forests. Lignocellulosic bioenergy crops are considered in nearly every IAM, and managed forestry for bioenergy is often excluded due to sustainability issues (Daioglou et al., 2020). Sugarcane cultivation is a key contributor to modern bioenergy supply (Ramirez Camargo et al.). It cannot be assumed that BECCS biomass supply will be coming from forest plantations or claim that this is the most conservative land use estimate, please revise text and calculations. There is a need to go down to country-level detail and identify the most likely feedstock, and maybe also do some different scenarios with varying feedstocks (residues, energy crops, etc.). I think data from Li et al. (2018) or Li et al. (2020) could be useful.	We agree with the reviewer that BECCS pledges are unlikely to be met by forest feedstocks, and we state clearly in the paper that we are only using this as a proxy removal factor, which we expect would have a similar rate to bioenergy feedstocks. However, we take the reviewer's point that this is an inadequate solution. At the same time, we do not agree that we should be modelling the variability for BECCS land area requirements depending on different feedstock options, or investigating what is most likely at a country level. We have already made extensive revisions to accommodate reviewer 1's concerns that these pledges should not be interpreted as reality and it is a key message of our paper that government pledges, if they are to be taken seriously, should include the expected land area required for implementation of the various CDR options they include. To avoid conveying a message that standing forests are likely to or should be used as

Daioglou, V., Rose, S.K., Bauer, N. et al. Bioenergy technologies in long-run climate change mitigation: results from the EMF-33 study. *Climatic Change* 163, 1603–1620 (2020). <https://doi.org/10.1007/s10584-020-02799-y>

Hanssen, S.V., Daioglou, V., Steinmann, Z.J.N. et al. Biomass residues as twenty-first century bioenergy feedstock—a comparison of eight integrated assessment models. *Climatic Change* 163, 1569–1586 (2020). <https://doi.org/10.1007/s10584-019-02539-x>

Li, W., Ciais, P., Stehfest, E., van Vuuren, D., Popp, A., Arneth, A., Di Fulvio, F., Doelman, J., Humpeönder, F., Harper, A. B., Park, T., Makowski, D., Havlik, P., Obersteiner, M., Wang, J., Krause, A., and Liu, W.: Mapping the yields of lignocellulosic bioenergy crops from observations at the global scale, *Earth Syst. Sci. Data*, 12, 789–804, <https://doi.org/10.5194/essd-12-789-2020>, 2020.

Li, W., Ciais, P., Makowski, D. et al. A global yield dataset for major lignocellulosic bioenergy crops based on field measurements. *Sci Data* 5, 180169 (2018). <https://doi.org/10.1038/sdata.2018.169>

Ramirez Camargo, L., Castro, G., Gruber, K. et al. Pathway to a land-neutral expansion of Brazilian renewable fuel production. *Nat Commun* 13, 3157 (2022). <https://doi.org/10.1038/s41467-022-30850-2>

BECCS feedstocks, we believe the best solution here is one which represents an average global value for BECCS capture rates, noting the wide variability based on feedstock as well as yields, conversion efficiencies, resource input, etc. After investigating several sources, we have decided to use a mean value for 2030-2050 from the LPJ-Guess model (Krause et al 2019).

This value changes the removal factor from between 11-13 tCO₂/ha/yr for the 5 countries for which we have quantified results for BECCS to 10.1 tCO₂/ha/year, increasing the total land area needed for BECCS from 74 to 82 million hectares.

We do not assume any of this feedstock is met with residues as analysis suggests residue availability equivalent to existing bioenergy demand (55EJ/yr) (Hanssen et al 2020). We have only identified BECCS in NDCs, which also include bioenergy demand in energy sector targets, and so we assume existing and energy-sector only demand is met through residues.

Finally, this does not represent a significant change to results. As we detail in the response to the next comment, our results are robust and the main conclusions stay the same across different variations, such as if we use a global average removal factor. Applying different bioenergy uptake assumptions could halve or double the land area we calculate for BECCS, and so we note in the paper the need for governments to provide more information about how they intend to achieve their BECCS mitigation pledges.

Hanssen, Steef V., Vassilis Daioglou, Zoran J. N. Steinmann, Stefan Frank, Alexander Popp, Thierry Brunelle, Pekka Lauri, Tomoko Hasegawa, Mark A. J. Huijbregts, and Detlef P. Van Vuuren. “Biomass Residues as Twenty-First Century Bioenergy Feedstock—a Comparison of Eight Integrated Assessment Models.” *Climatic Change* 163, no. 3 (December 2020): 1569–86. <https://doi.org/10.1007/s10584-019-02539-x>.

	Krause, A., V. Haverd, B. Poulter, P. Anthoni, B. Quesada, A. Rammig, and A. Arneeth. "Multimodel Analysis of Future Land Use and Climate Change Impacts on Ecosystem Functioning." Earth's Future 7, no. 7 (July 2019): 833–51. https://doi.org/10.1029/2018EF001123.
2. Belowground carbon/soil carbon dynamics are not mentioned at all, although it seems like Harris et al. considered this. I would expect to see a quantification of the soil carbon implications of the land area pledged for CDR, or at least, a solid discussion of what might be expected under different climatic conditions/forest types. In general, I would expect that relative to cropland both afforestation and bioenergy crops may enhance soil carbon stocks, although effects may be heterogenic. I include a couple of references that may help, although I am sure that there are other papers out there. Bell, S. M., Barriocanal, C., Terrer, C., & Rosell-Melé, A. (2020). Management opportunities for soil carbon sequestration following agricultural land abandonment. Environmental Science & Policy, 108, 104-111. https://doi.org/10.1016/j.envsci.2020.03.018 Cook-Patton, S.C., Leavitt, S.M., Gibbs, D. et al. Mapping carbon accumulation potential from global natural forest regrowth. Nature 585, 545–550 (2020). https://doi.org/10.1038/s41586-020-2686-x Qin, Z., Dunn, J. B., Kwon, H., Mueller, S., & Wander, M. M. (2016). Soil carbon sequestration and land use change associated with biofuel production: empirical evidence. Gcb Bioenergy, 8(1), 66-80. https://doi.org/10.1111/gcbb.12237 Ledo, A., Smith, P., Zerihun, A., Whitaker, J., Vicente-Vicente, J. L., Qin, Z., ... & Hillier, J. (2020). Changes in soil organic carbon under perennial crops. Global change biology, 26(7), 4158-4168. https://doi.org/10.1111/gcb.15120	We thank the reviewer for this comment, and we have reviewed the literature in terms of which carbon pools are included in removal factors. Harris et al. use only above ground removal factors for forests remaining forests in accordance with IPCC guidance of no change in belowground biomass. Below ground biomass increments are applied to aboveground removal factors for mangroves, plantations and young secondary forests. For our study, including a removal factor for these activities where the pledge type is emissions (rather than direct or indirect pledges) would apply to only 45 million ha (approx 5% of our results). On this basis, we have included a sensitivity analysis in our methodology discussion to show the difference in land area if below-ground biomass increment was included for the activities of mangroves, silvopasture, agroforestry and new forests, which decreases the total land area by 8.7 million ha. This now sits alongside the discussion of the sensitivity analysis we conducted to show the difference if we assumed a global average removal factor for all activities, which results in an increase in land area of 125 million ha, showing that the assumptions around activity types, biomes and removal factors constrain the land area calculations, but are not a key driver of results (given that just under half of the results are from direct area pledges). Both sensitivity analyses are now further explained in the SI, as well as being included in the online methods. Soil carbon in agricultural croplands is highly uncertain and the simulation of soil-carbon response to land use change varies across

	models. Soil carbon is increasingly subject to reversal in a warming climate, and the inclusion of this carbon pool would introduce even greater uncertainty to the results. See: Krause, Andreas, Thomas A. M. Pugh, Anita D. Bayer, Wei Li, Felix Leung, Alberte Bondeau, Jonathan C. Doelman, et al. "Large Uncertainty in Carbon Uptake Potential of Land-Based Climate-Change Mitigation Efforts." Global Change Biology 24, no. 7 (July 2018): 3025–38. https://doi.org/10.1111/gcb.14144. Viscarra Rossel, R. A., M. Zhang, T. Behrens, and R. Webster. "A Warming Climate Will Make Australian Soil a Net Emitter of Atmospheric CO₂." Npj Climate and Atmospheric Science 7, no. 1 (March 26, 2024): 79. https://doi.org/10.1038/s41612-024-00619-z.
3. Biophysical effects of land-based negative emission technologies will affect the performance of solutions. This must be highlighted, and the argument that it does not fit anywhere in the paper does not hold. Bioenergy crops have been associated with a cooling effect relative to cropland, although effects will vary based on location (see e.g., Wang et al. (2021), Wang et al. (2023), Muri (2018)). Afforestation in the tropics has been associated with a cooling effect, whilst for higher latitudes, reforestation warms the winter climate (Windisch et al., 2021). I think especially the latter is important for Russia's NDCs. Wang, J., Li, W., Ciais, P. et al. Global cooling induced by biophysical effects of bioenergy crop cultivation. Nat Commun 12, 7255 (2021). https://doi.org/10.1038/s41467-021-27520-0 Wang, J., Ciais, P., Gasser, T., Chang, J., Tian, H., Zhao, Z., ... & Li, W. (2023). Temperature Changes Induced by Biogeochemical and Biophysical Effects of Bioenergy Crop Cultivation. Environmental Science & Technology, 57(6), 2474-2483. Muri, H. (2018). The role of large—scale BECCS in the pursuit of the 1.5 C target: an Earth system	We thank the reviewer for this comment. We are familiar with the literature on biophysical effects of land use change and have reviewed the paper again in an effort to include a discussion on this. The paper focuses on the area of land that would be required to meet climate pledges, not the efficacy of these solutions. However, we have added a sentence on biophysical effects to the third paragraph of the discussion, where we discuss the impact of reforestation and restoration on local tenure and livelihoods.

model perspective. Environmental Research Letters, 13(4), 044010. Windisch, M. G., Davin, E. L., & Seneviratne, S. I. (2021). Prioritizing forestation based on biogeochemical and local biogeophysical impacts. Nature Climate Change, 11(10), 867–871. https://doi.org/10.1038/s41558-021-01161-z	
4. Should address the need for infrastructure for long-term CO2 storage somewhere for the cases of DACCS and BECCS. Rosa et al. highlights developing projects in Europe.' Rosa, L., Sanchez, D. L., & Mazzotti, M. (2021). Assessment of carbon dioxide removal potential via BECCS in a carbon-neutral Europe. Energy & Environmental Science, 14(5), 3086-3097. https://doi.org/10.1039/D1EE00642H	We view this as outside the scope of the paper. Our objective is to quantify the land area that would be required if governments were to implement CDR in climate pledges, not to assess the broader (and non land related) feasibility of this.
5. I suggest to put the NDC area pledges even stronger into the context of land use projections in 1.5C scenarios from integrated assessment. IIASAs AR6 database offers detailed data on specific scenarios that could be used to compare NDCs with future land use change for different combinations of Shared Socio-economic Pathways with Representative Concentration Pathways. Also, comparing with geospatial land use projections could offer valuable insights (e.g., Chen et al. (2020) or Hurtt et al. 2020)). https://data.ece.iiasa.ac.at/ar6/ Chen, M., Vernon, C.R., Graham, N.T. et al. Global land use for 2015–2100 at 0.05° resolution under diverse socioeconomic and climate scenarios. Sci Data 7, 320 (2020). https://doi.org/10.1038/s41597-020-00669-x Hurtt, George C., et al. "Harmonization of global land use change and management for the period 850–2100 (LUH2) for CMIP6." Geoscientific Model Development 13.11 (2020): 5425-5464. https://doi.org/10.5194/gmd-13-5425-2020 Data: https://luh.umd.edu/	Again, we feel that this is outside the scope of our paper, as we aim to assess government climate pledges and we feel that our discussion of how these compare to AR6 scenarios ranges is adequate, without comparing these to individual scenarios. We also note that the prominent 'State of CDR' report compares government commitments to different 1.5C mitigation scenarios and we do not see the value of repeating this. See: Smith et al 2023, State of CDR https://doi.org/10.1038/s41558-024-01984-6
6. I am wondering if the need for policy instruments to support CDR should be highlighted even stronger as a means to tighten the gap between pledges and actual deployment. See e.g., Wähling et al. for the case of BECCS.	It is not an objective of our paper to tighten the gap between pledges and deployment. As we note in the introduction, the objective is to make clear the land area required if this scale of CDR were to be deployed, something

Wähling, L. S., Fridahl, M., Heimann, T., & Merk, C. (2023). The sequence matters: Expert opinions on policy mechanisms for bioenergy with carbon capture and storage. Energy Research & Social Science, 103, 103215. https://doi.org/10.1016/j.erss.2023.103215	that governments have not made clear to date in their pledges. We also made revisions in response to the first round of review comments to clarify that the NDCs are aspirational targets and not to be interpreted as realistic future projections.
7. I cannot see that the Nabuur paper has been referenced, although claimed so in the rebuttal. I agree with reviewer 1's comment and with the message of Nabuurs et al. (2023) that carbon fluxes from unmanaged forests should ideally be reported, and that this point should be discussed somewhere. Nabuurs, GJ., Ciais, P., Grassi, G. et al. Reporting carbon fluxes from unmanaged forest. Commun Earth Environ 4, 337 (2023). https://doi.org/10.1038/s43247-023-01005-y	We thank the reviewer for noticing this. Not including the Nabuurs et al reference was an oversight. We have now added it into the discussion (at page 9, line 4).
P.2 Lines 18-21. Could also point out that global warming is a driver of biodiversity loss and that land-based climate change mitigation through afforestation or BECCS may help reduce impacts on biodiversity relative to a future with weaker mitigation efforts (see Iordan et al. (2023) and Hanssen et al. (2022)). Iordan, Cristina-Maria, et al. "Spatially and taxonomically explicit characterisation factors for greenhouse gas emission impacts on biodiversity." Resources, Conservation and Recycling 198 (2023): 107159. https://doi.org/10.1016/j.resconrec.2023.107159 Hanssen, S. V., Steinmann, Z. J., Daioglou, V., Čengić, M., Van Vuuren, D. P., & Huijbregts, M. A. (2022). Global implications of crop-based bioenergy with carbon capture and storage for terrestrial vertebrate biodiversity. GCB Bioenergy, 14(3), 307-321. https://doi.org/10.1111/gcbb.12911	We have revised the text on page 2 to clarify that climate change also has implications for biodiversity loss.
P.9 Lines 9-13. This is spot on. Meeting such land use changes at local levels would require major change in policies and local socio-technical conditions, they must be supportive enough, this seems challenging (see Næss et al. (2024)). Næss, J. S., Henriksen, I. M., & Skjølsvold, T. M. (2024). Bridging quantitative and qualitative science for BECCS in abandoned croplands.	We thank the reviewer for this comment. To stay within the scope of the paper (which is not to advise on policies and implementation, but to caution against the exaggerated expectations of current NDCs), we believe the important thing to highlight is the magnitude of land use transformation rates.

Earth's Future, 12, e2023EF003849. https://doi.org/10.1029/2023EF003849	
P.9 Lines 25-43. I think Russia’s CDR area pledge should be put in context with historical cropland abandonment (see Lesiv. et al.). In general, abandonment is widespread around the globe, and either letting this land regrow or converting it to bioenergy production/BECCS offers a good CDR potential (see Gvein et al.). Lesiv, M., Schepaschenko, D., Moltchanova, E. et al. Spatial distribution of arable and abandoned land across former Soviet Union countries. Sci Data 5, 180056 (2018). https://doi.org/10.1038/sdata.2018.56 Gvein, M.H., Hu, X., Næss, J.S. et al. Potential of land-based climate change mitigation strategies on abandoned cropland. Commun Earth Environ 4, 39 (2023). https://doi.org/10.1038/s43247-023-00696-7	We have added a sentence that around a tenth of the required land for Russia’s pledge could come from abandoned cropland as estimated by Lesiv et al. (2018). We are less inclined to accept that there are widespread abandoned cropland globally. While Gvein et al. do identify 98 Mha of abandoned cropland, most appear to be in former Soviet Union countries and a good amount - as the paper also recognises - are in biodiverse and water scarce areas and is therefore likely unsuitable for BECCS.
P.14 Lines 11-12. Can you be more specific and inform if this is dedicated planting of trees or natural regrowth, or both?	‘Young secondary forests’ refers to both tree planting and natural regrowth. The key difference from plantations is that young secondary are local mixed species. We have revised the sentence to make this clearer.
Table S1. Although Harris et al. seem convincing, some of these carbon dioxide removal factors may seem high to me. It is somewhat unclear if natural regrowth is relied on or if it also involved tree planting. Cook-Patton et al. provides an overview of natural regrowth rates in different climate zones, and they seem somewhat lower. I think some more comparisons with other literature could be beneficial to provide an indication of if there is a variation in reported values or not.	There is variation in removal factors, but those from Harris et al., as well as the silvopasture and agroforestry removal factors we used, are based on IPCC default values, which in the absence of location-specific data we believe is the best approach. We also note that our results are not entirely reliant on removal factors, as a significant area of land results from direct or indirect pledges, and point the reviewer to our sensitivity analyses which show an 11% increase in land area if a uniform global removal factor was applied, and an 0.002% decrease in total land area if below ground biomass is included in removal factors.

Response to review

Thank you for the improvements made to the manuscript. The soil carbon sensitivity for selected activities was appreciated. I note that several of my comments were not acted upon, and in some cases solid justification was given. However, other aspects still requires revisions. The results presented here are very similar to the Land Gap Report. Several figures contain similar data. More care should be taken to provide citations to the Land Gap report where there is clear overlap, including for data in individual figures. Examples include figure 2 and figure 3 in the manuscript. E.g., figure 1 in the 2023 land gap report that shows land required for CDR in national climate pledges, which is a different way to visualize the data shown in Figure 3 in the submitted manuscript. It even states some of the same/similar country-shares (for example, Russia 35%, US 12%, Saudi 20%). Figure 3 in the 2023 land gap report is like Figure 2 in the submitted manuscript (both figures include data on land requirements and number of countries). https://landgap.org/downloads/2022/Land-Gap-Report_FINAL.pdf https://landgap.org/downloads/2023/Land-Gap-Report_2023-Briefing_FINAL.pdf	This paper builds on and extends the analysis of the Land Gap report, as noted in our letter to the editor accompanying the first submission. We stated: “This manuscript is based on the 2022 Land Gap Report, but extends the previous analysis by including all climate pledges (NDCs and LT-LEDs) submitted until the end of 2022, and includes an uncertainty analysis on pledges that do not directly state land area. This article includes temporal and geographical distribution of pledges which were not discussed in the Land Gap Report.” This was also noted by the first reviewers, with reviewer #2 stating: “As a publication based on the analysis of the "Land Gap Report", it provides a timely contribution to the scientific literature, esp. in the context of next NDCs due in 2025”. Given the time that has elapsed since submission, we have now updated the analysis to include all pledges made until the end of 2023. We have added a reference to the Land Gap report on page 3, lines 3-4. We have also referenced specific line numbers below where changes are made in response to reviewer comments. Additional edits in the manuscript are to reduce word count. All page references given here are for the clean version of the manuscript.
BECCS calculations still have major improvement potential. It is too simple for a paper addressing land requirements of CDR. Using a global average BECCS carbon removal	We have updated our approach to BECCS calculations based on this reviewer’s recommendations. More detail is provided below.

rate based on LPJ-Guess to quantify national-level BECCS land requirements does not follow state-of-the-art methods, considering all the spatial yield data that is available from multiple sources (including LPJ-Guess). On top of that, comes opportunities to utilize land-free bioenergy feedstocks (see for example, Wu et al.). I agree that a change in approach is unlikely to affect the main conclusion of the paper (e.g. over-reliance of land in pledges) considering the importance of reforestation and restoration, but it will affect sub-results and sub-findings that support the main conclusion including all BECCS results. Currently, the national-level results and land requirements for the five countries with BECCS pledges has low value, perhaps especially important for the US pledge. It still needs revision. Wu, F., Pfenninger, S., & Muller, A. (2024). Land-free bioenergy from circular agroecology—a diverse option space and trade-offs. Environmental Research Letters, 19(4), 044044.	The value of the national level findings is primarily limited by the lack of information provided in NDCs and long-term strategies, not by the different methodological approaches that could be taken to analysing these pledges. It is not the objective of this paper to provide detailed or accurate insights into national level land-use strategies, but rather to reveal what information is contained in national climate pledges relating to land use, and what that looks like at a global aggregate. To this end (and based on recommendations from reviewer #2) we state on page 3, lines 6-11 that: “While the information given by countries in their climate pledges is of insufficient detail to provide accurate assessments of the amount of land that would be required for CDR, and the pledges themselves cannot be taken as precise descriptions of what will happen in the future, our analysis provides a first estimate of the implications for global land pressure of national climate pledges. More transparency and consistency in country pledges would facilitate future analysis...”
Using nation-specific yields would already help. As noted below, it makes more sense to consider second generation energy crops in 2050 than first generation. And a quantitative basis for the discussions on implications of different bioenergy feedstocks on land requirements should be provided.	We have now adjusted the analysis to use nation-specific yields (using the Li et al 2020 dataset with 2nd generation energy crops). We have responded in more detail below regarding how this dataset was used, and the discussion added to the paper regarding the quantitative implications of different bioenergy feedstock and other conversions efficiencies for land area requirements
I note that recently published research has provided scenario analysis of the land use	Thank you for pointing us to this paper. The difference in quantified land

implications of the NDCs for a SSP2 scenario (SSP2-NDC). I think that the insights provided there considering how NDCs affects future land use is important and merits a mention here. E.g., in SSP2-NDC increased forest cover towards 2060 comes at the expense of rangeland and other natural land (nonforested ecosystems, shrublands, deserts). Also note that the net land use change towards forests and BECCS seems lower than the land requirements quantified here.

Humpenöder, Florian, et al. "Food matters: Dietary shifts increase the feasibility of 1.5° C pathways in line with the Paris Agreement." *Science Advances* 10.13 (2024): eadj3832. <https://doi.org/10.1126/sciadv.adj3832>

requirements between this paper and our work is because Humpenöder quantify NDCs, while our analysis includes 2050 pledges, in addition to NDCs. One of the key findings from our analysis is that land-reliant pledges significantly scale up *after* 2030, i.e., in 2050 pledges. This explains the difference in results compared to a paper that looks at NDCs and we have changed the title of our paper to make this finding clearer, emphasising the over-reliance on land use is in *net-zero* pledges, not in NDCs.

We have included a reference to Humpenöder et al in terms of the decreased need for CDR and hence pressure on land achieved through dietary shifts (see p.5, line 14), but a direct comparison between our results and Humpenöder et al is not appropriate.

P.2 Lines 21-24, and several other places in different sections in the manuscript addresses/discusses impacts on biodiversity. CDR may help hinder irreversible biodiversity losses caused by increasing temperatures that leads species niche limits to exceed (see Trisos et al.). The manuscript does a very good job in highlighting the sustainability risks of land use change. I think there is wide agreement that CDR should not be used as an excuse to avoid emission cuts (e.g., p9 lines 41-47). However, the lack of a discussion of the wider benefits of achieved land-based mitigation from CDR on the same sustainability indicators makes the communication too one sided (see Hirata et al. for a thorough analysis of land-based CDR biodiversity implications). I'd like to see some more nuance in the discussions.

Hirata, A., Ohashi, H., Hasegawa, T. et al. The choice of land-based climate change mitigation measures influences future global biodiversity loss. *Commun Earth Environ* 5, 259 (2024). <https://doi.org/10.1038/s43247-024-01433-4>

We have revised the language about biodiversity on p. 2, lines 20-23 to specify that CDR efforts, if successful in mitigating climate change, could reduce biodiversity loss. We do maintain that ample research highlights that large-scale CDR poses significant risks to biodiversity (which is also recognised and evidenced in the Hirata-paper). We also note that Hirata compares with a no-mitigation baseline scenario. If mitigation is equated with CDR, then obviously CDR will have positive effects on biodiversity, yet we note that mitigation is more than just CDR. We therefore have not revised the text elsewhere, where we highlight the risks CDR poses to biodiversity.

We thank you for pointing us in the direction of Trisos et al., which we have included. We have also added a reference to Hirata, who note that land-use change associated with CDR can have negative regional impacts on biodiversity (p. 2, lines 24-25).

Trisos, C.H., Merow, C. & Pigot, A.L. The projected timing of abrupt ecological disruption from climate change. Nature 580, 496–501 (2020). https://doi.org/10.1038/s41586-020-2189-9	
P.5 Lines 18-26. The simple approach of using a global average rate for BECCS is inconsistent with the attempt to capture some locality that is done for restoration and reforestation. It also disregards a lot of work done to spatially model energy crops over the last couple of decades, including using the LPJ family of models (LPJ-Guess mentioned and cited in-text). The approach used here does not follow state-of-the-art. It is something that could be easily improved, and that also should be improved. Chosen approach might not be expected to change your main conclusion (e.g., over-reliance on land in climate pledges), but it does affect sub-results and sub-findings that supports the totality. A key example is United States and results in Figure 3. I re-iterate the need to improve the modelling of bioenergy crops and to capture effects of locality for the five countries with BECCS pledges.	These climate biomes are a standard way to present forest emissions factors and removals factors, (see eg. Harris et al) and are based on IPCC guidance for LULUCF accounting. No such guidance exists for bioenergy or BECCS, hence there are no standardised or accepted proxy values to represent BECCS, as has been developed for forests. Such values are likely to be developed via the upcoming IPCC methodology report on CDR, expected to be published in 2027: https://unfccc.int/sites/default/files/resource/IPCC_TFI_AR7%20overview_2024.pdf However, we have taken the reviewer’s recommendation to use country-specific yield values for bioenergy, as detailed below.
P5. Lines 19-22. As a more concrete example, you could for example point to that Switzerland and UK could meet a share of their BECCS pledges with no land requirements by implementing CCS in incinerators (waste-to-energy, also true for other countries). See Rosa et al. for a quantification, I suggest comparing emission pledges to country-specific potentials. Rosa, L., Sanchez, D. L., & Mazzotti, M. (2021). Assessment of carbon dioxide removal potential via BECCS in a carbon-neutral Europe. Energy & Environmental Science, 14(5), 3086-3097.	We have added a reference to this paper in terms of the potential for a portion of BECCS demands to be met via waste, see p. 5, lines 24. However, we reiterate our point that not all bioenergy demand is for BECCS, particularly over the next few decades (see Egerer et al, preprint). Our aim is to estimate the land demand if countries were to meet BECCS pledges via dedicated crops, and we are transparent about the variance in BECCS assumptions (page 5, lines 4-5). The paper from Rosa et al, as recommended by the reviewer, also states that it is unlikely the full potential for waste-to-energy utilization will be realised, and we note that the pledges from UK and Switzerland alone would assume 50% of this waste energy

	potential. While the reviewer is only proposing a proportion of these potentials would be met via waste, this illustrates the scale of bioenergy pledges (from only 2 European countries) vs. the scale of waste potential (see p. 10, lines 20-22). As we state at p. 6, lines 1-2, countries do not provide information on how BECCS pledges will be met. There are a multitude of choices and assumptions in determining bioenergy feedstocks, we have made this clearer at p. 6, lines 3-5, but we believe the approach taken here gives a reasonable mid-range estimate, as we further explain below. Egerer, Sabine, Stefanie Falk, Dorothea Mayer, Tobias Nützel, Wolfgang Obermeier, and Julia Pongratz. "How to Measure the Efficiency of Terrestrial Carbon Dioxide Removal Methods," May 22, 2024. https://doi.org/10.5194/egusphere-2024-1451.
P.5 Lines 27-30. It is unclear why bioenergy should be separated into energy sector pledges and BECCS pledges. Cannot these pledges perfectly align, as BECCS is a multi-functional process which both produces energy and delivers negative emissions? In some cases, as for wastes (for example, CCS in incinerators or biomethane production with CCS), it may even serve at least three functions, including waste treatment, energy production, and delivery of negative emissions. Note that there are 119 ongoing CCS projects under development in Europe, several which involve incinerators. Levina, E., Gerrits, B., & Blanchard, M. (2023). CCS in Europe – Regional Overview. Global CCS Institute. https://www.globalccsinstitute.com/resources/publications-reports-research/ccs-in-europe-regional-overview/	We have not separated the pledges, this is how they are presented in NDCs, with many countries including bioenergy for energy use without including BECCS. Hence we expect the global bioenergy demand to not be only driven by BECCS pledges in the near-term. While BECCS and energy pledges may perfectly align in theory, we remind the reviewer that the objective of this paper is not to build an idealised scenario with state of the art or most efficient choices modelled at every step, but to represent the potential land demand of existing national climate pledges. Many countries pledge to use bioenergy without including BECCS, and so we assume there will be demand on bioenergy feedstocks beyond what we have quantified here for BECCS.

	This assumption is illustrated in the recent pre-print from Egerer et al, which argues that not all bioenergy crops can be expected to be used for BECCS in the near term. They model a scenario where BECCS appropriation of energy crops increases over coming decades, meaning that by 2050, capture rates are still relatively low compared to later in the century. Hence, we believe it remains a reasonable assumption that waste feedstocks are in coming decades used to meet other bioenergy demands. Egerer, Sabine, Stefanie Falk, Dorothea Mayer, Tobias Nützel, Wolfgang Obermeier, and Julia Pongratz. "How to Measure the Efficiency of Terrestrial Carbon Dioxide Removal Methods," May 22, 2024. https://doi.org/10.5194/egusphere-2024-1451.
P5 lines 23-24 and P11 lines 14-15. Does Krause et al. (2019) that you rely on for bioenergy capture rates from LPJ-Guess specifically address bioenergy and first-generation energy crops (edible feedstocks)? I see no clear indication of it, neither in the article, in their SI, or in the data published on Figshare. Did I miss something, or is this the wrong citation (is it really Krause et al. (2018)?)? I guess Krause et al. (2019) quantifies NPP of some plant functional types, but these were probably not parameterized as typical bioenergy feedstocks. Please clarify how Krause et al. was used. Krause, A., V. Haverd, B. Poulter, P. Anthoni, B. Quesada, A. Rammig, and A. Arneth. "Multimodel Analysis of Future Land Use and Climate Change Impacts on Ecosystem Functioning." Earth's Future 7, no. 7 (July 2019): 833–51. https://doi.org/10.1029/2018EF001123. Krause, A. et al. Large uncertainty in carbon	Model variables are described in Krause et al 2019. PFTs are C3 and C4 cereal crops (wheat and maize). The global mean value was provided by the authors directly, however, we have now changed to Li et al 2020 based on this reviewer's recommendations.

uptake potential of land-based climate-3 change mitigation efforts. Glob Change Biol 24, 3025–3038 (2018).	
As all the BECCS pledges are for 2050 (according to land calculator), it is surprising to rely on first-generation crops and not second generation. Daioglou et al. highlights that lignocellulosic crop (including miscanthus, willow, and eucalyptus) dominates future bioenergy supply in IMAGE scenarios, whilst edible crops play a minor role (see fig. 7). The same also happens in other models. Also, unclear why irrigated crops were chosen and not rain-fed, especially since water scarcity risks were previously highlighted. Daioglou, Vassilis, et al. "Integrated assessment of biomass supply and demand in climate change mitigation scenarios." Global Environmental Change 54 (2019): 88-101. https://doi.org/10.1016/j.gloenvcha.2018.11.012	Most ESMs in CMIP6 do not distinguish second-generation bioenergy crops and other crops yet (Harper et al., 2018), with PFT representing 2nd generation bioenergy only more recently being included across multiple DGVMs, in part informed by studies such as Li et al 2018, but yields are only one aspect of bioenergy values (see extended discussion below). We now use Li et al 2020 for bioenergy yield values, which includes both irrigated and rainfed 2nd generation energy crops. Daioglou et al 2019 note that meeting 20% of final energy demand for 1.5C targets through bioenergy “can only be achieved without extreme levels land use change if agricultural yields improve significantly and effective land zoning is implemented, and future technologies such as 2nd generation crops and CCS are utilised”. We have included a reference to this at p. 5, lines 15-17 and p. 6, lines 3-5 to highlight that land pressure from CDR is dependent on many choices which have not yet been made apparent in national climate pledges.
As an example on the importance of crop types, Li et al. predicts with machine learning techniques a global mean yield of 16.3 tDM ha⁻¹ yr⁻¹ for lignocellulosic crops, which may correspond to about 8 tC ha⁻¹ yr⁻¹ (or sequestered 30tCO₂ ha⁻¹ yr⁻¹) harvested. If 90% of this is captured in a thermal power plant, then this is nearly 3x higher removal rate than the value used in your work (10.1 tCO₂ ha⁻¹ yr⁻¹, Table S1) from Krause et al 2019 and way outside the ranges used in the uncertainty analysis. Li, Wei, et al. "Mapping the yields of	We are now using Li et al 2020 for country specific yield values. See below for extended response to this point.

lignocellulosic bioenergy crops from observations at the global scale." Earth System Science Data 12.2 (2020): 789-804. https://doi.org/10.5194/essd-12-789-2020	
---	--

Extended response to selection of bioenergy crop yields and conversion efficiency for BECCS:

There are large uncertainties in BECCS per hectare capture rates, with results varying by at least a factor of three, as the reviewer has pointed out. Most papers on the topic explicitly note a high uncertainty in BECCS potentials. There is also a divergence between the type of deployment modelled scenarios suggest is necessary to prevent large scale land conversions and food security issues (such as suggested by Daioglou et al 2019), and what governments are proposing in their climate mitigation strategies. Our aim is to highlight this divergence, and not to make high efficiency assumptions on behalf of government policies where choices have not yet been made explicit.

For BECCS capture rates, yields are only one part of the equation, with many other process-based losses. While IAMs include these process-based losses where they occur, and use conversion efficiencies to represent crop harvest, DGVMs tend to use conversion efficiencies to capture a multitude of process-based losses. Vaughan et al 2018 discuss the full life-cycle analysis of BECCS, showing the various processes that need to be captured by conversion efficiency, concluding that conversion losses are commonly 40-60%.

The reviewer uses a 90% conversion efficiency in their example here, which we find to be extremely high. IAMs capture "the key process and land use change emissions that can influence the net CO₂ removed by a BECCS system, but this is not explicitly quantified in a single value" (Vaughan et al 2018), meaning that taking the figures for BECCS sequestration per hectare from IAMs may be misleading. In DGVMs, the yield numbers don't account for differences in bioenergy area across grid cells, where not all fractions of a gridcell contain energy crops (see Krause et al 2018, Egerer et al (pre-print)).

Hence, yield values alone cannot be used to determine land area required for BECCS without capturing other process-based losses. Observed yields across the literature are higher than modelled yields for the reasons outlined above (with process-based losses factored into modelled yields to varying degrees). Conversion efficiency varies across studies, tending to be lower in DGVMs (where biomass harvest efficiency number incorporates other process-based inefficiencies) and higher in IAMs (where emissions from other process-based inefficiencies are captured elsewhere in the model). Hence taking yield values from observational field plot sites warrants a conservative conversion efficiency, given the values are being directly applied rather than modelled.

We thank the reviewer for the suggestion of the Li et al 2020 dataset, but would like to point out some reasons why we previously declined to use this dataset as a proxy for bioenergy capture rates:

- Li et al 2020 note that the purpose of their dataset is to be used by IAM or process-based models as yield inputs. These models then include conversion and supply chain losses.
- Li et al 2020 detail that most of the observations in the training data are from small-scale experimental trials with management practices rather than real farmers' fields (data taken from Li et al., 2018a). Hence the yields might not be repeated under large-scale farm conditions.
- Li et al note that low yields of less than 4 t DM ha⁻¹ yr⁻¹ are much more common in the observations than the observed maximal yields in the dataset (observations to be found in Li et al., 2018a) and observe that "it may need more water and nutrients in order to sustain the high yields."
- Littleton et al 2020 note that the difference between modelled and observed results is clearly illustrated in the southern United States, where modelled yields are as much as 20 t DM ha⁻¹ yr⁻¹ higher than observations, and the largest BECCS pledge is from the United States.

However, in the absence of a unified dataset from multiple DGVMs producing regionally disaggregated data based on 2nd generation bioenergy crops, which would provide the ideal underlying values for our purpose, we have used the yield values from Li et al 2020 as suggested by the reviewer. We have coupled this with a 60% conversion efficiency rate, which is at the upper end of Vaughan et al 2018 suggested process losses. To show that the resulting values still fall at the upper end of available global means we have compiled a table below for easy comparison.

Observed and modelled mean yields (highlighting dataset and conversion efficiency used in this revision):

reference	Yield (t DM/ha)	conversion efficiency	t/CO ₂ /ha removal factor
Krause et al 2018 (LPJ-GUESS)	11.1 - mean	80%	16
Krause et al 2019 (LPJ-GUESS)	13.7 - mean	60%	10.11
Harper et al 2018 (JULES)	10.4 - mean	60%	11.45
Harper et al 2018 (IMAGE)	15.8 - mean	80%	23
Littleton et al 2020 (JULES)	12.4 - mean	60%	13.65
Egerer et al – pre-print (JSBACH)	12.7 - median	60%	13.98
Li et al 2018 (observational)	11.5 - median	If 60%	12.66
Li et al 2018 (observational)	11.5 - median	If 80%	16.88

Li et al 2018 (modelled)	14.3 - mean	If 80%	20.99
Li et al 2020 (simulated)	16.3 - median	If 60%	17.94
Li et al 2020 (simulated)	16.3 - median	If 80%	24

Note – all conversion values given are those used in studies, apart from Li et al 2018 and Li et al 2020 which constitute datasets as input to scenarios (ie: no conversion rates provided). Hence, we show options for conversion efficiency to use with Li et al datasets. A 60% conversion efficiency brings these within the mid-range of other studies, compared to the top end of the range with 80% conversion efficiency. Orange row indicates value now used in this study.

Egerer, Sabine, Stefanie Falk, Dorothea Mayer, Tobias Nützel, Wolfgang Obermeier, and Julia Pongratz. "How to Measure the Efficiency of Terrestrial Carbon Dioxide Removal Methods," May 22, 2024. <https://doi.org/10.5194/egusphere-2024-1451>.

Harper, Anna B., Tom Powell, Peter M. Cox, Joanna House, Chris Huntingford, Timothy M. Lenton, Stephen Sitch, et al. "Land-Use Emissions Play a Critical Role in Land-Based Mitigation for Paris Climate Targets." *Nature Communications* 9, no. 1 (December 2018). <https://doi.org/10.1038/s41467-018-05340-z>.

Krause, Andreas, Thomas A. M. Pugh, Anita D. Bayer, Wei Li, Felix Leung, Alberta Bondeau, Jonathan C. Doelman, et al. "Large Uncertainty in Carbon Uptake Potential of Land-Based Climate-Change Mitigation Efforts." *Global Change Biology* 24, no. 7 (July 2018): 3025–38. <https://doi.org/10.1111/gcb.14144>.

Krause, A., V. Haverd, B. Poulter, P. Anthoni, B. Quesada, A. Rammig, and A. Arneeth. "Multimodel Analysis of Future Land Use and Climate Change Impacts on Ecosystem Functioning." *Earth's Future* 7, no. 7 (July 2019): 833–51. <https://doi.org/10.1029/2018EF001123>.

Li, Wei, Natasha MacBean, Philippe Ciais, Pierre Defourny, Céline Lamarche, Sophie Bontemps, Richard A. Houghton, and Shushi Peng. "Gross and Net Land Cover Changes in the Main Plant Functional Types Derived from the Annual ESA CCI Land Cover Maps (1992–2015)." *Earth System Science Data* 10, no. 1 (January 30, 2018): 219–34. <https://doi.org/10.5194/essd-10-219-2018>.

Li, Wei, et al. "Mapping the yields of lignocellulosic bioenergy crops from observations at the global scale." *Earth System Science Data* 12.2 (2020): 789–804. <https://doi.org/10.5194/essd-12-789-2020>

Littleton, Emma W., Anna B. Harper, Naomi E. Vaughan, Rebecca J. Oliver, Maria Carolina Duran-Rojas, and Timothy M. Lenton. "JULES-BE: Representation of Bioenergy Crops and Harvesting in the Joint UK Land Environment Simulator Vn5.1." *Geoscientific Model*

Development 13, no. 3 (March 11, 2020): 1123–36. <https://doi.org/10.5194/gmd-13-1123-2020>.

Vaughan, Naomi E, Clair Gough, Sarah Mander, Emma W Littleton, Andrew Welfle, David E H J Gernaat, and Detlef P Van Vuuren. “Evaluating the Use of Biomass Energy with Carbon Capture and Storage in Low Emission Scenarios.” *Environmental Research Letters* 13, no. 4 (April 1, 2018): 044014. <https://doi.org/10.1088/1748-9326/aaaa02>.

Response to review continued:

P5, lines 25-26. And IAMs show lower yield values for second generation bioenergy crops in comparison with the random forest model from Li et al. cited above (see fig 5). IAM yield maps (IMAGE, Magpie, GLOBIOM) also underestimate yields relative to field observations (see fig 6, Li et al.). Are you sure it is right to “give the impression” that IAMs rely on too high energy crop yields? Also, text says there is more information in SI, but I did not find any	As explained above, yield values between observational data and modelled approaches are not directly comparable. However, we have deleted this sentence.
P.5 line 25. IAM abbreviation not spelled out previously, should remove abbreviation and spell out. Might need to spend a few words describing what these models are as well.	This sentence (and abbreviation) has now been deleted.
P6, lines 11.13. According to Bluwstein & Cavanagh, the land acquisition peak in 2011 was about 12.4 Mha yr⁻¹, not 7 Mha yr⁻¹. See fig. 10. Also, should specify that this is Global South only, and not worldwide. If a mean is taken over a period around 2011, then 7Mha could be right, but then please specify the year range. Strongly recommend comparing against remotely sensed data in addition, see for example the Hilda+ paper (Winkler et al.). It shows annual changes in the 2000s between 6-11 Mha yr⁻¹ (see fig3). Bluwstein, J., & Cavanagh, C. (2022). Rescaling the land rush? Global political ecologies of land use and cover change in key scenario archetypes for achieving the 1.5 °C Paris agreement target. The Journal of Peasant Studies, 50(1), 262–294. https://doi.org/10.1080/03066150.2022.2125386	Thanks for this comment. Good point that we are less clear on this. We had presented an average of what Bluwstein & Cavanagh (p280) call a 'spike' of transactions over the period 2007-14. We have now clarified that in the text (p. 7, lines 23-26). Thanks also for mentioning Winkler et al. (2021). This paper uses a variety of remote sensing data to come up with global land use change estimates over time. It shows annual changes since 1960 of ~70 Mha yr⁻¹ with changes peaking at >80 Mha yr⁻¹ in the mid-2000s (Fig. 3). However, these are gross changes across a number of land use categories, whereas we're looking at net change towards

Winkler, K., Fuchs, R., Rounsevell, M. et al. Global land use changes are four times greater than previously estimated. Nat Commun 12, 2501 (2021). <https://doi.org/10.1038/s41467-021-22702-2>

forest/plantation, so a direct comparison would not make sense. Fig. 4 in Winkler et al. (2021) illustrates that the net changes across different land use change categories are much smaller than the gross changes, yet neither the paper nor the supplementary information files contain estimates of annual net change rates. As a response to your follow-up question, we do believe that the comparison with land transactions is relevant, as we are here interested in thinking about the challenges of directing net land use change through governance interventions and the associated possible risks for land governance and rural livelihoods. For these reasons, we have decided not to use Winkler et al. (2021), but rather to clarify our use of Bluwstein and Cavanagh (2022) further. Where the old text read "For example, the global land rush of the 2000s, which was seen as a great threat to small scale farmers' land tenure security and livelihoods, saw no more than seven million ha being transacted per year.", the new text now reads "For example, over the period 2007-14, which was the most intensive period of what has been dubbed 'the global land rush', an average of seven million ha was transacted per year in the Global South. This development was seen as a great threat to small scale farmers' land tenure security and livelihoods."

Furthermore, to clarify that we are here specifically focusing on the governance and livelihood challenges associated with large-scale directed land use changes, we have tweaked the language of the

	last concluding paragraph of this section (p.8, lines 8-11). The old text read: "Hence, our analysis suggests the rates of land-use change already included in national climate pledges, at 13 million ha per year, are unprecedented from a historical perspective, and comparable to the average rates of land transformation assumed in global modelled scenarios by mid-century - scenarios that have raised significant concerns within the scientific community exactly over their vast consequences for land use." The new reads: "Hence, our analysis suggests that the rate of direct land use change for carbon removal included in national climate pledges, at 13 million ha per year, is unprecedented from a historical perspective. Furthermore, it is comparable to the average rates of land transformation assumed in global modelled scenarios by mid-century that have raised significant concerns within the scientific community exactly over their vast consequences for land use, governance and rural livelihoods."
A follow up question to this. How good of a proxy is land acquisition / transactions as a land-use change indicator? My guess is that it is far from perfect. Re-iterate the need to consider remote sensing products.	We have responded to this above, but would also like to point out that a follow-up study (Lay et al 2021) found that by 2020 between 30-73% of the transacted area was converted to agricultural production (as we state at p. 7, lines 27-29). Given the land transactions we are referring to occurred in the 2000s, land-use in 2020 is a reliable indicator. Lay, J. et al. Taking Stock of the Global Land Rush: Few Development Benefits, Many Human and Environmental Risks.

	Analytical Report III. https://boris.unibe.ch/156861/ (2021) doi:10.48350/156861.
P7. Line 5. 322 million ha is the mean (not specified)? Also, across what scenarios? SSPs in combination with RCP-1.9?	322 million ha is the median. This is specified in the final sentence of the paragraph, but we have now added it into the first sentence as well. The scenarios are described as “Modelled pathways that limit warming to 1.5°C with no or limited overshoot”. This is clearly specified in AR6 to refer to C1 (SSP1-1.9) scenarios, and the reference is given to AR6 WGIII where this information can be readily found. However, given the interdisciplinary nature of the journal, we believe our description is sufficient, and anyone familiar with C1 scenarios would identify them by our description. Adding technical detail on scenario labels would not improve reader understanding of the statement, which we have tried to keep accessible, while still specific.
P7 Line 8. Provide the full range here as well? The complete feasibility space is important. These pathways should not be viewed as a statistical sample (see Huppmann et al., box 1). Huppmann, D., Rogelj, J., Kriegler, E. et al. A new scenario resource for integrated 1.5 °C research. Nature Clim Change 8, 1027–1030 (2018). https://doi.org/10.1038/s41558-018-0317-4	As suggested by Huppmann et al 2018, we have referred to the full range from the scenario set, not only the median. However, we take the reviewer’s point that the pathways should not be viewed as a statistical sample, and we have added this to the manuscript, along with a reference to Huppmann et al 2018 at page 7, lines 7-8).
P.7 lines 46-48. Refers to the grain-for-green programme? Perhaps, it should be mentioned directly.	We refer here not only to the ‘grain-for-green’ or what is also called ‘Conversion of Cropland to Forest Program’ program, but to the larger and longer history of China’s efforts at tree planting and forest restoration, including, for instance, the earlier, large efforts at

	shelterbelt plantings. A good overview of these different efforts is provided in chapter 2 of this book, https://doi.org/10.17528/cifor/002116.
P.9 Lines 18-19. This depends on where forest expansion happens. Too unspecific, please spend some more words to explain.	We have expanded this sentence (p. 11, lines 16-18, however, we note this is a complex issue which could be the subject of an entire paper, and is not the subject of this paper.
P9. Line21-22. Should differentiate between increases in forest cover for carbon removal and increases in land area for energy crops. This statement is only right for forest cover, not for energy crops (see page 7, lines 3-9). These two land uses do not serve the same function(BECCS can be used both for energy and CDR).	The reviewer seems to be referring to this statement: “There are well founded concerns that land use change on this scale would be particularly pronounced in the Global South, where historical trends of pasture and cropland expansion would need to be reversed, leading to an absolute reduction in these land uses”. We don’t agree with the reviewer that this statement would apply only to increases in forest cover and not to increases in energy crops. Energy crops do not indicate the same land use as pasture and cropland, which is why they are noted as a separate category, e.g. in IAMs.
Page 9. Lines 41-43. Land requirements of future food production or biodiversity impacts have not been modelled here quantitatively, so how robust is this conclusion? As it stands now, it seems like qualitative speculation with insufficient support. Note that in contrast Xu et al. highlights that delaying implementation of land-based mitigation measures may threaten food security due to feedback loops on global warming. Some comparisons with quantitative studies may be needed to make this conclusion (planetary boundaries? Hirata et al. cited above?). Should have a look at Humpenöder et al. fig 6 showing some expected loss of natural area in SSP2-NDC. Xu, S., Wang, R., Gasser, T. et al. Delayed use of bioenergy crops might threaten climate and food	We acknowledge that we haven’t modelled the possible impacts on food production and biodiversity of the changes in land use set out in national climate pledges. However, there is ample support in a wide literature for the claims that (i) taking land out of agricultural production in the global South may compromise food security for local populations and (ii) afforestation and tree planting rarely benefits biodiversity, and is far less beneficial for biodiversity than avoiding deforestation. References

security. Nature 609, 299–306 (2022). https://doi.org/10.1038/s41586-022-05055-8 Humpenöder, Florian, et al. "Food matters: Dietary shifts increase the feasibility of 1.5° C pathways in line with the Paris Agreement." Science Advances 10.13 (2024): eadj3832. https://doi.org/10.1126/sciadv.adj3832 Also note that many scenarios with major expansion of land-based mitigation involves agricultural intensification and a decrease in pasture area to free land for mitigation purposes. The latter relies on dietary shifts. This strategy can help avoid sustainability impacts.	at p. 8, line 11 and p. 11 line 12 substantiate these points. On the point of how delayed implementation of land-based mitigation measures could threaten food security through feedback loops on global warming, we want to note that we are clearly not calling for delayed mitigation - also not for delayed land-based mitigation. We are pointing at the fact that national climate pledges appear to rely heavily on large-scale transformations of land use with possible risky side-effects, potentially at the expense of safer and more immediate possibilities for mitigation, i.e. reducing fossil fuel reliance. In response to your comment, we have rewritten this paragraph to make our points more clearly (see p.11, lines 41-45).
P10. Lines 3-5. The justification for this statement ("unrealistic targets") is to my impression primarily based on previous land use change rates. This argument needs to be repeated here again for more clarity.	Given this is the conclusion, we don't feel it is necessary to repeat previous arguments made in the paper that support the conclusion, especially for a short-format paper of 3000 word limit. The statement 'unrealistic targets' (now at p. 12, line 4) is preceded by the words 'what appear to be', indicating this statement is to be interpreted on the basis of claims made in our paper.
P.11 Line 35. Specify Table S2?	Thank you. We have now specified tables in all references to SI where relevant.
P11. Lines 30-37. How have you accounted for uncertainty in FAO reported forest extent for indirect pledges? Is it included in the error bar	Uncertainties in FAO land cover data are a well known problem, but the forest cover data has improved

shown in Fig 1b? This did not become clear to me after checking the land calculator either. FAO data on forests generally seems somewhat disputed, Lesiv et al. and Bastin et al. both show higher forest cover compared to FAO FRA data.

Lesiv, M., Schepaschenko, D., Buchhorn, M. et al. Global forest management data for 2015 at a 100 m resolution. *Sci Data* 9, 199 (2022). <https://doi.org/10.1038/s41597-022-01332-3> ‘

Bastin, Jean-François, et al. "The extent of forest in dryland biomes." *Science* 356.6338 (2017): 635-638.

in recent years as FAOSTAT incorporates forest data from the latest FRA publication, which is systematically carried out and up to date. However, uncertainty estimates are not provided with FAO values, and so we have not included any. We have revised the paper at p. 13, line 50.

We have reviewed the use of FAO forest area and now only rely on this for 2 data rows (and a 3rd using agricultural land area), totalling 7,069 ha. A further 7 rows are based on country land area, but country estimates of their total land area are more certain. (this can be seen by filtering the Land Gap calculator for indirect pledges (column P), and the column H shows what these indirect pledges are based on - FAO forest land or land area, or Crowther et al tree density). The lack of uncertainty values for these data rows will not noticeably affect the results.

There is value in FAO data and it is widely used in analysis, particularly for its status as recognised official country statistics. This plays an important role in countries accepting the results of work based on their own statistical information. There are advantages and disadvantages between the use of FAO and satellite data, as discussed in Houghton & Castanho 2023, and both remain valid approaches.

Houghton, Richard A., and Andrea Castanho. "Annual Emissions of Carbon from Land Use, Land-Use Change, and Forestry from 1850 to 2020." *Earth System Science Data* 15, no. 5 (May 23, 2023): 2025–54.

	https://doi.org/10.5194/essd-15-2025-2023 .
Fig 1. A land use change for bioenergy/BECCS is not equal to reforestation. Please improve visualization and differentiate. I did not find the explanation of indirect pledges very clear in the caption, perhaps spending some more words there to describe it could help. I suggest to also put some more information in the graph by stacking the bars and separating into continents or similar, although this might be a more subjective recommendation.	Thank you for this feedback. We have re-conceptualized this figure to show the contribution of BECCS to land use change, as well as the extra land area added through conditional pledges. Land areas for other activities are provided in Table 2. Whether land use change is the same for reforestation versus bioenergy crops really depends on the original land-use. We are drawing a broad proxy here that land-use change carries greater sustainability risks, and that greater area of land required for climate change mitigation increases the likelihood of land use change from natural ecosystems, in particular grasslands.
Fig. 4. Same comment as before, that bioenergy/BECCS is not reforestation (at least, affecting UK). Caption is not informative enough to understand it for a reader with limited time skimming through. Suggest providing some more detail.	We have revised the figure caption (and labels) to refer to land use change / no land use change rather than restoration / reforestation, given that bioenergy crops are likely to require a land use change but not to forests.
For BECCS, it is unclear what carbon losses are assumed throughout the supply chain, what carbon capture efficiency is assumed, and how you deal with any supply chain leakages of CO₂. Capture efficiency will vary a lot between conversion pathways such as bioelectricity, biofuel, biomethane, etc. This directly affects BECCS land use.	We have discussed this in detail above where we explain how this is usually treated in process-based and IA models, and how we have factored this into the Li et al 2020 yield data using a 60% conversion efficiency rate.
All the BECCS pledges are for 2050, but it is unclear what background climate was used to produce LPJ-GUESS yields. Expected effects of climate change	This is one of the reasons we would prefer to use yield rates from a DGVM, where taking values from 2020-2050, as we did in our

on yields and consequentially the land requirement should be highlighted.	previous revision, captures the expected effects of climate change on yields. Such effects are not captured in country specific yield data from Li et al, as noted by the authors. We have made a reference to this in the paper, p. 13, lines 13-14. Also note that the BECCS pledges are to be achieved by 2050, not in 2050, implying they will need to start very soon.
The same question regarding impacts of climate change on forest growth. Harris et al. studied the past period of 2001-2019. How should climate change be expected to affect removal factors and quantified future land area requirements from emission pledges? Especially important for Russia's major emission-based pledge in 2050?	We have added a discussion and references to this at p. 13, lines 15-19.
As a final note, I would have appreciated if you could have helped me as a reviewer (with limited time available) out by referring in the rebuttal specifically to the lines were you made changes in the manuscript or by quoting the change in the rebuttal. As a reviewer, my key interest is to see what improvements were made in the manuscript, not only the response to comments.	We apologise for this oversight, and have now included page and line numbers for all revisions discussed in this response to review.

Response to review

The authors have been very responsive to my comments, both by improving the paper and providing clarifications. I include some further reflections from my side that should be straightforward to address. Line numbers refer to the clean manuscript version. I am overall happy that the comment on bioenergy feedstock, yields, and efficiencies was taken seriously. There is now more transparency considering the relative importance of yields and what is here termed conversion efficiency (combination of carbon capture efficiencies, and other losses). The choice of a 60% conversion rate for the main results shown comes across as reasonable, as this is somewhere in between what may be expected across different conversion pathways such as for bioelectricity, biofuels, biogas, etc. Thank you for the increased clarity.	We thank the reviewer for their thorough comments. We appreciate that the reviewer agrees with our choice of the 60% conversion rate.
I found the table that was included in the rebuttal on removal factors to be very useful as it exemplifies the effect of feedstock, chosen models/observations and conversion efficiency, and I recommend to include it also in the supplement to ensure coverage of a larger feasibility space. Perhaps with additional columns in the table indicating feedstock and effects on quantified land use (hectare). It is important in order to understand how results are affected by subjective choices. In fact, should probably even include another lower conversion efficiency in the range of 40-50% representing a liquid biofuel pathway (such as Fischer-Tropsch diesel), see for example Hanssen et al. Table S2 for reviewed CCS efficiencies. Likewise, high-end performing bioelectricity pathways achieving 80-90% conversion efficiency is a part of the feasibility space and should in my opinion be shown as a minimum in an SI. It is definitely not about making excuses for government	We thank the reviewer for this comment, but we do not think it is appropriate to include this table in the SI, or to add additional columns to it. The table was compiled to make it clear to the reviewer why we thought 60% was a reasonable conversion efficiency rate. The reason we do not wish to include the table in the SI is that bioenergy, and the different conversion efficiencies and other process losses over different bioenergy pathways is not the subject of our paper. In particular, conversion to biofuels is not relevant to this paper as we are only dealing with BECCS. The table is also incomplete, as many more studies could be added, and not all of the data is published. For example, the global mean from Krause et al 2018 was provided to us directly by

policies, but rather to inform about the feasibility space and being rigorous.

Hanssen, S.V., Daioglou, V., Steinmann, Z.J.N. et al. The climate change mitigation potential of bioenergy with carbon capture and storage. Nat. Clim. Chang. 10, 1023–1029 (2020). <https://doi.org/10.1038/s41558-020-0885-y>

the authors. We strongly feel that a table such as this, and the topic of conversion efficiencies and other assumptions around BECCS could be the subject of a paper in its own right, and indeed, these issues have been explored in many papers (i.e., Harper et al., 2018, Vaughan et al. 2018, and the pre-print from Egerer et al contains a similar table).

We have extensively referenced this and other literature from the bioenergy community, including several papers the reviewer has recommended.

We agree with the reviewer that results are affected by subjective choices, and we have added additional explanation to the SI and the methods section to highlight this:

In the methods (page 13, lines 17-19):
“Subjective choices around conversion efficiency, which varies from 40% to 90% across studies, can significantly impact the results in terms of land area required, and hence the perceived feasibility space.”

In the SI:
“We note that the key processes and land use change emissions that can influence the net CO₂ removed by a BECCS system is not easily quantified in a single value, such as conversion efficiency, and this is treated differently across different approaches to quantifying BECCS uptake (Vaughan et al 2018), with potentially large differences in results.”

As also stated by the authors, 2nd generation energy crops have recently been

We contacted several authors working with DGVM models and 2nd generation

implemented and parameterized in multiple DGVMs, fully coupled ESMs and similar frameworks. Thus, if country-level or biome yield data for energy crops from DGVMs or similar can be obtained I would support their use (but not a requirement). See for example:

Stenzel, F., Greve, P., Lucht, W. et al. Irrigation of biomass plantations may globally increase water stress more than climate change. *Nat Commun* 12, 1512 (2021). <https://doi.org/10.1038/s41467-021-21640-3>

Upgraded LPJmL5 version https://www.negemproject.eu/wp-content/uploads/2021/06/NEGEM_D3.1.pdf

Cheng, Yanyan, et al. "A bioenergy-focused versus a reforestation-focused mitigation pathway yields disparate carbon storage and climate responses." *Proceedings of the National Academy of Sciences* 121.7 (2024): e2306775121.

Li, W., Yue, C., Ciais, P., Chang, J., Goll, D., Zhu, D., Peng, S., and Jornet-Puig, A.: ORCHIDEE-MICT-BIOENERGY: an attempt to represent the production of lignocellulosic crops for bioenergy in a global vegetation model, *Geosci. Model Dev.*, 11, 2249–2272, <https://doi.org/10.5194/gmd-11-2249-2018>, 2018.

Ai, Z., Hanasaki, N., Heck, V., Hasegawa, T., and Fujimori, S.: Simulating second-generation herbaceous bioenergy crop yield using the global hydrological model H08 (v.bio1), *Geosci. Model Dev.*, 13, 6077–6092, <https://doi.org/10.5194/gmd-13-6077-2020>, 2020.

Melnikova, I., Ciais, P., Tanaka, K. et al. Relative benefits of allocating land to bioenergy crops and forests vary by region. *Commun Earth Environ* 4, 230

bioenergy crops, and as yet none of these have made country level yield data available. We are aware of several projects where this will be available over the coming years, at which point we would use this in any future iterations of this work.

We cannot include data beyond what the modellers have yet made available.

We agree that regional studies vary across models, and some may perform better than JULES, but we have not used values from JULES in our results.

(2023). <https://doi.org/10.1038/s43247-023-00866-7>

Also, it can be observed that several, or most, of the studies do compare predicted yields with observations. Regional performance varies across models. Some of them may perform better in the US than JULES (Littleton et al.).

P6. lines 7-9. I read the response, but it still did not become clear why this disaggregation in energy and BECCS pledges should be used as a justification to assume that other bioenergy demand outside of BECCS should be met with wastes and residues. While I agree that non-BECCS land usage for bioenergy doesn't need to be modelled here, the statement draws an artificial boundary between biomass feedstock for bioenergy and biomass feedstock for BECCS that does not necessarily exist considering BECCS multifunctionality. CCS can be implemented in a variety of conversion pathways relying on different feedstocks. I suggest to delete the second part of the sentence, e.g. "assuming that bioenergy demand outside of BECCS could be met with wastes and residues". Also suggest to instead simply restate that the focus is on CDR and not energy as the reason to not model bioenergy land requirements (or an alternative could be to just delete lines 7-9 altogether). Otherwise, I find the text already added on the potential contribution from wastes in Europe sufficient (and useful!).

We thank the reviewer for this suggestion, as we did not intend to draw an artificial boundary between biomass feedstock for bioenergy and biomass feedstock for BECCS. We have amended the sentence as suggested (to focus on CDR), but have left the statement that we did not address bioenergy demand outside of BECCS so readers are aware those additional demands still exist. (page 6, lines 7-9).

P11. lines 20-21. I need to repeat a comment for new consideration, as I don't think the message came through and the wrong sentences were quoted. I'll try again, being more clear. Here in page 11 it is stated: "It is alarming that the extent of land required for CDR in government climate pledges already tracks against the upper end of mid-century scenario expectations."

Thank you for clarifying this comment. We have amended the sentence to say: "It is alarming that the extent of land required for CDR in government climate pledges already tracks against the upper end of mid-century scenario expectations for reforestation, with only 5 pledges made for BECCS to date."

I note from Table 2 that land use change for reforestation is quantified as 450 Mha and for BECCS as 61 Mha.

Then in page 8, the following is stated:

"Modelled pathways that limit warming to 1.5°C with no or limited overshoot show increases in forest cover for carbon removal of 322 million ha (median, with a range of -67 to 890 million ha)¹⁶. Many of these pathways also include large amounts of energy cropland area, to supply biomass for bioenergy and BECCS, with 199 (median, 56-482 range) million ha by 2050."

450 Mha of land use change for reforestation is indeed above median and in the upper range/half in 1.5C pathways (although, still half of 890 Mha in the far high-end). However, 61 Mha for BECCS is far below median and bordering the lower end of the 56-482 Mha range (although range also include land use for bioenergy in addition to BECCS, and shares between the two vary across models (see Daioglou et al. (2020) for EMF-33 energy results, fig1)).

Considering that BECCS land requirements is only about one quarter of the median in 1.5C pathways and borders the low end of the range, I therefore propose to differentiate between land use change for increased forest cover and energy crops in p11 lines 20-21. Otherwise, the statement seems misleading.

Daioglou, V., Rose, S.K., Bauer, N. et al. Bioenergy technologies in long-run climate change mitigation: results from the EMF-33 study. Climatic Change 163, 1603–1620 (2020). <https://doi.org/10.1007/s10584-020-02799-y>

(page 11, lines 20-22).

fig 1. BECCS was separated, but now it is unclear what "conditional" and "unconditional" pledges in the legend means. While these two terms may be

Conditional and unconditional pledges refers to the way developing countries distinguish action that is

established elsewhere, they are not used anywhere else in this manuscript (I tried to search). Adding an explanation here would be helpful. Also, with current figure design, seems like BECCS fits neither conditional or unconditional? Is that correct, or should BECCS also be separated (BECCS-conditional and BECCS-unconditional, or only one of the two)?	contingent on additional climate finance in their NDCs. This distinction does not apply to BECCS, as only developing countries make conditional pledges, and only developed countries have included BECCS pledges. We have added an explanation to the figure label: “Additional actions pledged by developed countries that are contingent on climate finance are shown as conditional pledges.”
fig 3. I noticed that the source data given in column B in ("455217_3_data_set_9406687_shhr29.xlsx", sheet "Figure 3 share of global land") differs from the labels given in the figure next to country names. Fix?	Thank you for pointing this out, this has been fixed so that column B includes the correct values.
P.13 lines 11-13 refers to the SI for details on bioenergy country yield calculations. In the SI, in addition to providing some country-numbers (Table S1) the following is written: "The yield uptake rates and SD for bioenergy were taken from Li et al 2020, with conversion efficiencies applied to yield uptake following Vaughan et al 2018". This short explanation is not very transparent and makes the results difficult to replicate. How did you go from gridded data from Li et al. to country-specific yields? Not sure what was done, but a decent proxy could be to use gridded data of current cropland cover to filter and weight yield data. Also, what does the standard deviation represent, is it calculated based on spatial yield variability within the country? Do you include conversion efficiency in the SD as well? My impression is no, and if so, I really think you need to include in the SI the table from the rebuttal describing effects of yields and conversion efficiencies to better cover the feasibility space.	To convert gridded data in Li et al to country specific yields we used a GIS software (QGIS) and applied country specific polygons. Based on ‘best crop estimate’ in each pixel, the software provides mean, min, max, variance and SD for each polygon (country). This method resulted in a global mean of 16.12 compared to 16.4 reported by Li et al., which we consider close given variation in software pixels within country boundaries. SD was calculated as variation on the mean for each country, hence this would indicate spatial variation based on mean value in each pixel. Conversion efficiency of 60% is applied both to the mean and the SD. We have added a sentence to the SI describing the above approach. We would like to remind the reviewer that it is not our aim here to develop country-specific bioenergy yield

	values, but to use what is available from the bioenergy literature. It is not an objective of this paper to advance the bioenergy literature, rather to draw from it. For this reason, as well as the ones specified above, we decline to include the table describing effects of yields and conversion efficiencies as this is something that should come from within the bioenergy community.
p.11 41-47. Some of these claims are still very strong with limited quantitative support from the analysis that was done. It has indeed been shown here that some large countries that rely on petroleum production have made major CDR pledges with associated land use. I am however less convinced that it has really been shown that pledges push mitigation burden onto the land sector instead of fossil fuel phase out. Although it probably is right, there is no quantitative basis here provided on emission cuts supporting the statement. Same regarding biodiversity and food security beyond land use indicators. I note that some text was shuffled around but that the message seems the exact same as in the previous manuscript version. The point made that taking land out of production in Global South may compromise food security for local populations is valid considering the geographical distribution of pledges shown, but perhaps it should be written in directly to support the claim. In contrast, I feel like the claims made in the abstract is better balanced.	In order to avoid making claims that are not supported by our quantitative results, we have rephrased this section make the conclusion that pledges embed unrealistic expectations on land, and we hypothesize, based on the geographical distribution, that some countries push the mitigation burden onto land: “Our results show that these pledges embed another layer of insufficiency in that they embed unrealistic expectations of the land sector. The geographical distribution of land claims in our dataset illustrates that especially major fossil producers with large land areas rely on land-based CDR. This pattern could suggest that countries push the mitigation burden onto the land sector rather than phasing out emissions from fossil fuels and land-use change.” (page 11, lines 42-47).